# Machine-learning model selection and parameter estimation from kinetic data of complex first-order reaction systems

**László Zimányi**[1], **Áron Sipos**[1], **Ferenc Sarlós**[1], **Rita Nagypál**[1,2], **Géza I. Groma**[1]*

**1** Institute of Biophysics, Biological Research Centre, Eötvös Loránd Research Network, Szeged, Hungary,
**2** Doctoral School of Physics, University of Szeged, Szeged, Hungary

* groma.geza@brc.hu

**Data Availability Statement:** All the data as well as the software applied in this study is available at https://github.com/groma-geza/FOkin.

## Abstract

Dealing with a system of first-order reactions is a recurrent issue in chemometrics, especially in the analysis of data obtained by spectroscopic methods applied on complex biological systems. We argue that global multiexponential fitting, the still common way to solve such problems, has serious weaknesses compared to contemporary methods of sparse modeling. Combining the advantages of group lasso and elastic net—the statistical methods proven to be very powerful in other areas—we created an optimization problem tunable from very sparse to very dense distribution over a large pre-defined grid of time constants, fitting both simulated and experimental multiwavelength spectroscopic data with high computational efficiency. We found that the optimal values of the tuning hyperparameters can be selected by a machine-learning algorithm based on a Bayesian optimization procedure, utilizing widely used or novel versions of cross-validation. The derived algorithm accurately recovered the true sparse kinetic parameters of an extremely complex simulated model of the bacteriorhodopsin photocycle, as well as the wide peak of hypothetical distributed kinetics in the presence of different noise levels. It also performed well in the analysis of the ultrafast experimental fluorescence kinetics data detected on the coenzyme FAD in a very wide logarithmic time window. We conclude that the primary application of the presented algorithms—implemented in available software—covers a wide area of studies on light-induced physical, chemical, and biological processes carried out with different spectroscopic methods. The demand for this kind of analysis is expected to soar due to the emerging ultrafast multidimensional infrared and electronic spectroscopic techniques that provide very large and complex datasets. In addition, simulations based on our methods could help in designing the technical parameters of future experiments for the verification of particular hypothetical models.

**Funding:** LZ, FS, NZ, GG; GINOP-2.3.2-15-2016-00001; Economic Development and Innovation Operative Programme of Hungary; https://www.palyazat.gov.hu/node/56577 LZ; K-124922; National Research, Development and Innovation Office of Hungary; https://nkfih.gov.hu AS; NKFIH PD-121170; National Research, Development and Innovation Office of Hungary; https://nkfih.gov.hu LZ, GG; 2018-1.2.1-NKP-2018-00009; National Research, Development and Innovation Office of Hungary; https://nkfih.gov.hu The funders had no role in study design, data collection and analysis, decision to publish, or preparation of the manuscript.

**Competing interests:** The authors have declared that no competing interests exist.

## Introduction

From classical flash photolysis [1–8] to recent methods of ultrafast time-resolved spectroscopy [9–14], light-induced kinetic studies—especially those carried out on macromolecules—face the challenge of analyzing a complex scheme of reactions. One reason for such complexity is related to the lengthy cascade of reactions initiated by photoexcitation. A typical example is the sequence of photointermediates of retinal proteins, including the complicated scheme of the bacteriorhodopsin (bR) [15] photocycle. Complexity is also attributed to the heterogeneity of the conformational states and/or the microenvironment of a chromophore studied with time-resolved fluorescence [9, 11] or transient absorption methods [14]. Regardless of the degree of complexity, in most of the above problems we can suppose that the individual steps of the reaction scheme are well approximated by first-order kinetics. In such cases, the problems can be addressed by standard methods designed to solve systems of first-order linear homogeneous differential equations [16–18].

Briefly, the temporal evolution of a given scheme of reaction components can be described by

$$\frac{d\mathbf{c}(t)}{dt} = \mathbf{K}\mathbf{c}(t), \tag{1}$$

where $\mathbf{c}(t)$ is an $n$-vector containing the concentrations of the components at time $t$ and $\mathbf{K}$ is known as the microscopic rate matrix. For $i \neq j$, $K_{i,j}$ represents the rate constant of the reaction from component $j$ to component $i$, and $K_{i,j}$ is the negative sum of the outward rates from component $i$. The solution of Eq (1) with appropriate initial conditions requires solution of the eigenvalue problem of $\mathbf{K}$. Constraints among the elements of the matrix ensure that its eigenvalues are always real [17]. If the eigenvalues are non-degenerate, as generally supposed in routine analyses, the solution of the differential equations can be expressed as a sum of exponential terms, with rate constants equal to the eigenvalues called macroscopic rate constants. Due to the coupling of the different components in Eq (1), an individual exponential term appearing in its solution—and observable for the experimentalist too—may belong to the time evolution of more than one component. If the data are obtained by a spectroscopy experiment, the spectral distribution of the amplitude of such a term is called decay-associated spectrum (DAS) or difference spectrum (DADS) [18].

In the light of the above possible coupling, because of the numerous unknown factors, it is impossible to achieve the most ambitious aim of determining the $\mathbf{K}$ matrix of a given reaction scheme by target analysis [18], i.e. only from temporal and spectrotemporal experimental data. Such an analysis requires repeating the experiment with various parameters (e.g. temperature and/or pH), building models on how the rate constants depend on these parameters and making assumptions on the spectra of the participating components [6, 19]. An interesting novel approach for handling this problem for a relatively simple set of light-induced reactions utilizes a deep learning network trained with synthetic time-dependent spectra [20]. Without a high amount of experimental data and a priori knowledge, the common approach is to determine the macroscopic rate constants and the corresponding DAS/DADS by global fitting with $n$ exponential terms [21]. Although this is a much simpler task, it still implies problems:

*P1* In many cases there is no well-established prior knowledge suggesting that all participating reactions are really of first order.

*P2* Experimental data often provide rather poor information because only a relatively low number of data points are available in a wide time range.

*P3* The number of components $n$ is not known in advance.

*P4* The nonlinear fit requires pre-estimation of all unknown parameters.

*P5* There is no guarantee of reaching the true global minimum, which may not be unique.

*P6* Exponential fitting is inherently an ill-posed problem: even low error input data generate high uncertainty in the estimated parameters [22, 23].

Most of these issues can be avoided if instead of discrete exponentials the predicted result is characterized by a distribution on a quasi-continuous space of time constants and the nonlinear regression problem is extended by a regularizing penalty term [11, 22–27]. In a recent paper [28] we argued to solve the problem

$$\text{minimize} \quad \frac{1}{2}\|\mathbf{b} - \mathbf{A}\mathbf{x}\|_2^2 + \lambda\|\mathbf{x}\|_1, \tag{2}$$

where $\mathbf{b}$ is the vector of the experimental data with a length of $m$, the element $b_i$ of which is taken at the time $t_i$, $\mathbf{A}$ is an $m \times n$ matrix—called the design (or measurement) matrix—with elements

$$A_{ij} = \exp\left(-t_i/\tau_j\right), \tag{3}$$

$\tau_j$ is an element of the $n$-vector $\boldsymbol{\tau}$, consisting of a series of pre-defined time constants, $\mathbf{x}(\boldsymbol{\tau})$ is the distribution to be determined, $\lambda$ is a positive hyperparameter, and the $L_1$ and (the not squared) $L_2$ norm of a vector $\mathbf{v}$ are defined as

$$\|\mathbf{v}\|_1 = \sum_{i=1}^{n} |v_i|, \quad \|\mathbf{v}\|_2 = \sqrt{\sum_{i=1}^{n} v_i^2}. \tag{4}$$

In the literature of signal processing, problem (2) is termed Basis Pursuit Denoising (BPDN) [29], while in statistics its widely used name is lasso for Least Absolute Selection and Shrinkage Operator [30]. Here we use the latter, more current acronym, but maintain the standard notations of signal processing [31]. The most important property of the lasso is that it not only guarantees a regularized solution $\mathbf{x}$, but also a sparse one [32], in accordance with the principle of parsimony, a fundamental rule in model selection [33]. Sparsity ensures a close connection to the original discrete exponential terms. Regularization and sparsity together minimize the appearance of invalid features in the solution due to noise, hence address *P6*. Since for a given problem (2) a fixed value of $\lambda$ unequivocally determines the number of peaks in the solution, *P3* is eliminated. In addition, the lasso is a convex—but not necessarily strictly convex—problem, which reduces the difficulties with *P4* and *P5*. Problem *P2* is partially handled by the fact that a sparse solution may be obtained even if $n \gg m$–the favorable condition to gain detailed information on the distribution $\mathbf{x}$–without introducing information into the solution other than that contained in the data themselves. Recently, lasso regularization has been applied for the analysis of time-resolved spectroscopic data in other laboratories, and it is an option in the PyLDM package [34, 35].

The aim of the present study is to exceed the capabilities of the simple lasso method in three main directions. First, we intend to enable it to analyze multidimensional kinetic data, taking into account the correlations among them. This requirement is obvious e.g. for spectroscopic data, where one expects that the kinetics at every wavelength can be characterized by the same set of time constants. Here we show that the problem can be solved by an extension of the lasso, called group lasso [31, 36]. Second, to target the yet unresolved problem *P1*,

we find that another extension, elastic net [37] with an additional hyperparameter, does a very good job in controlling the sparsity of the solution **x** continuously from a very low to a very high level. Finally, and most importantly, by applying the arsenal of modern statistics [38]–particularly cross-validation [39, 40] and Bayesian optimization [41, 42]–we have constructed a machine-learning system for completely automatic model selection. This task is equivalent to determining the value of the two hyperparameters exclusively from the data, corrupted by noise of an unknown level. The excellent performance of the above methods is demonstrated on a simulated dataset based on a rather complex model of the bR photocycle as well as on experimentally determined ultrafast fluorescence kinetic data taken on the coenzyme flavin adenine dinucleotide (FAD). The workflow for the sequence of algorithms is depicted in Scheme 1.

**Scheme 1**. **The workflow for building increasingly complex algorithms for accurate model selection**.

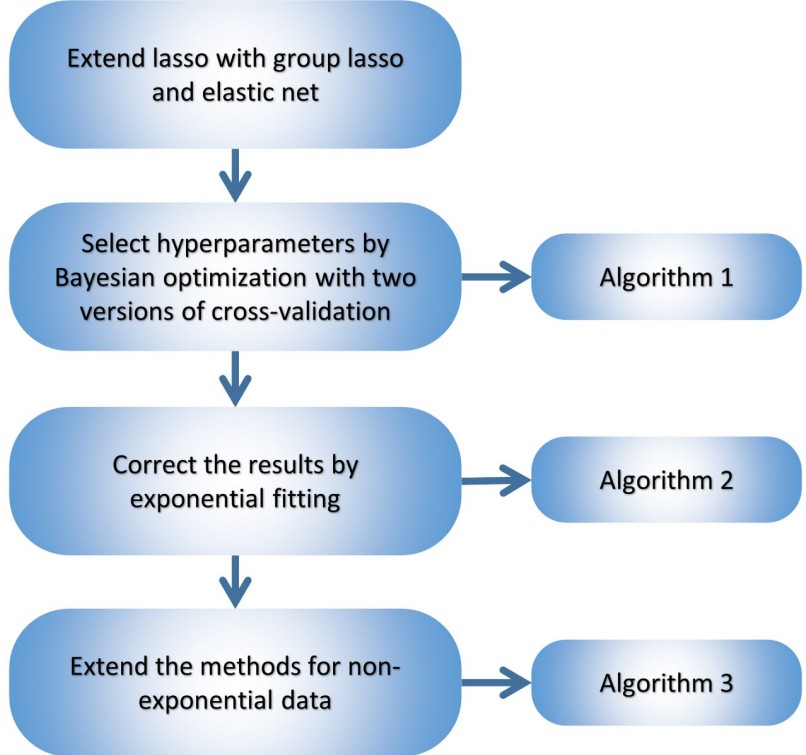

## Theoretical basis

In this section we briefly outline the statistical methods used in this work. For more details we recommend the excellent introductory books of Hastie and coworkers [32, 38].

### Methods for parameter estimation

The group lasso. The group lasso is an extension of the lasso defined in Eq (2) for the case when some groups of the elements of **x** are in correlation. Partitioning **x** into subvectors $\mathbf{x} = (\mathbf{x}_1, \ldots, \mathbf{x}_G)$, where the elements of any $\mathbf{x}_g$ form a correlated set of values, the group lasso

problem [36] is defined as

$$\text{minimize} \quad \frac{1}{2}\|\mathbf{b} - \mathbf{A}\mathbf{x}\|_2^2 + \lambda \sum_{g=1}^{G} \|\mathbf{x}_g\|_2. \tag{5}$$

This definition ensures that for any $g$ either all or none of the elements of $\mathbf{x}_g$ will be nonzero [32]. This property of the solution is similar to that obtainable by global multiexponential fitting. Obviously, if each subvector $\mathbf{x}_g$ is of length one, Eq (5) is equivalent to Eq (2).

In the special case when the kinetic data are obtained by a spectroscopic method, one expects a correlation among the elements of $\mathbf{x}$ corresponding to different wavelengths but the same value of the time constant. On the other hand, no such correlation is expected among the elements corresponding to different time constants. In this case, the above partitioning of $\mathbf{x}$ and the building of a giant design matrix related to that is rather inconvenient. Therefore, choosing a matrix form is a more natural representation. Supposing that the observations were taken at $p$ wavelengths defined in a vector $\mathbf{w} = (w_1, \ldots, w_p)$, the unknown distribution can be arranged in an $n \times p$ matrix $\mathbf{X}$, whose element $x_{jk}$ corresponds to time constant $\tau_j$ and wavelength $w_k$. Let $\mathbf{x}_{j,*}$ and $\mathbf{x}_{*,k}$ denote the $j^{th}$ row and $k^{th}$ column of $\mathbf{X}$, respectively. The kinetic data themselves are arranged in a set of vectors $\mathbf{b}_k$, $k = 1, \ldots, p$, corresponding to wavelength $w_k$. The design matrix $\mathbf{A}_k$ is defined individually for each value of $k$. This freedom is useful if the elements of the matrix cannot have the simple form as defined in Eq (3), typically when we need to take into account the convolution of the signal with the instrument response function of a measuring apparatus [14, 18]. For grouping across the individual elements in each row of matrix $\mathbf{X}$ one can set the group lasso problem as

$$\text{minimize} \quad \frac{1}{2}\sum_{k=1}^{p} \left\| \left(\mathbf{b}_k - \mathbf{A}_k\mathbf{x}_{*,k}\right) \right\|_2^2 + \lambda \sum_{j=1}^{n} \|\mathbf{x}_{j,*}\|_2. \tag{6}$$

Obviously, Eq (6) can be extended with other parameters, like additional spaces of wavelength in the case of multidimensional vibrational or electronic spectroscopy.

**The elastic net.**   The elastic net problem [37] combines $L_1$ and $L_2$ penalties in the form of

$$\text{minimize} \quad \frac{1}{2}\|\mathbf{b} - \mathbf{A}\mathbf{x}\|_2^2 + \lambda \left[ \frac{1}{2}(1 - \alpha)\|\mathbf{x}\|_2^2 + \alpha\|\mathbf{x}\|_1 \right], \tag{7}$$

where $\alpha \in [0, 1]$ is a second hyperparameter. Since the $L_2$ penalty alone does not induce sparsity in the solution, the variation of $\lambda$ and $\alpha$ in their entire range results in solutions varying from very dense to very sparse. One can expect that this property of the elastic net can handle problem *P1*. In addition, for $\alpha < 1$ the elastic net problem is strictly convex, eliminating problems *P4* and *P5*. Regularization with both $L_1$ and $L_2$ penalties was applied for the analysis of low-resolution NMR relaxation kinetic data [43–45].

To utilize the advantages of both the group lasso and the elastic net, in this study we use their combination in the form of

$$\text{minimize} \quad \frac{1}{2}\sum_{k=1}^{p} \left\| \left(\mathbf{b}_k - \mathbf{A}_k\mathbf{x}_{*,k}\right) \right\|_2^2 + \lambda \left[ \frac{1}{2}(1 - \alpha)\sum_{k=1}^{p} \|\mathbf{x}_{*,k}\|_2^2 + \alpha\sum_{j=1}^{n} \|\mathbf{x}_{j,*}\|_2 \right]. \tag{8}$$

In the sequel this optimization problem is referred to as the group elastic net problem (GENP). Since there are sudden changes in the objective function of Eq (8) if $\alpha$ approaches the

value 1, it is technically more appropriate to calculate with the variable $\omega = 1 - \alpha$ in a logarithmic scale, as we do in this study.

Solving the GENP with a given set of kinetic data at fixed values of $\lambda$ and $\omega$ provides an estimation for the values of **X**. To execute this task, one needs a model selection procedure to determine the value of these hyperparameters.

## Methods for model selection

**Cross-validation.** Despite its introduction in the 1970s [39, 40], to our best knowledge the method of cross-validation (CV) has not been applied in analyses outlined in the main section of the Introduction. For our purposes, CV must be considered in the context of a machine-learning procedure for model selection [38], which needs independent data for the training and testing phases. A CV procedure solves this task by randomly dividing the same dataset into training and testing data. The most common CV algorithm is the $k$-fold CV, which randomly sorts the whole dataset into $k$ subsets. During machine learning, $k$-1 sets are applied for learning, i.e. for estimating some model parameters by fitting to data in these sets. Then the remaining single set is applied for testing. In the testing phase, the data are compared to the values simulated by the parameters calculated in the learning phase and the mean square error (MSE) is calculated. The procedure is executed $k$ times, with each subset selected once for testing. A certain model is characterized by the mean of the MSE values obtained in each turn. Consequently, CV is a promising tool for model selection. Models with more free parameters typically result in a higher goodness of fit than those with fewer ones. However, this effect can be partially attributed to the overfitting of noisy data, leading to a superfluous complexity of the model. One can expect that if a set of models is characterized by the different values of one or more hyperparameters, the model selected at the minimum value of the mean MSE calculated by CV will yield an optimal trade-off between goodness of fit and complexity. Despite the popularity of $k$-fold CV in model selection, the theoretical justification of this expectation for models applying lasso for parameter estimation is unclear [46]. Moreover, practical simulation results indicate that for these types of models in the high-dimensional setting ($n \gg m$), $k$-fold CV tends to bias towards unjustified complexity [47].

In a recent paper [48] Feng and Yu point out that one possible reason for the above bias in $k$-fold CV is that across its different splits, the support of the solution **x** (the subset of components with nonzero values) can change. Averaging the MSEs over these misaligned structures is not justified. To deal with this problem the authors suggest a special version of leave-$n_v$-out CV, selecting repeatedly and randomly $n_c$ data from the whole dataset of length $n$ for learning and leaving out $n_v = n - n_c$ for testing [49]. The key point of their algorithm is that in the first step the support of **x** is determined by a penalized estimator like lasso, and CV is calculated with a restricted design matrix, leaving out columns not included in the support. In this restricted space no penalty is used; only the maximum likelihood estimator (the first term in Eq (2)) is applied. It is proven that under proper conditions this restricted leave-$n_v$-out CV (RCV($n_v$)) is consistent in hyperparameter selection, meaning that with $n \to \infty$ the probability of the selected model being the optimal one approaches 1. Obviously, this eliminates the bias problem of $k$-fold CV, as also justified by simulation results.

**Bayesian optimization.** The model selection method described above requires determining the minimum of a black-box function $f(\lambda, \omega)$, whose values can be calculated by executing a CV procedure at fixed pairs of the hyperparameters $\lambda$ and $\omega$. For lasso-like problems it is common to perform these calculations on a pre-defined path of $\lambda$ values. The main advantage

of this method is that a very effective algorithm for solving these problems—implemented e.g. in the popular glmnet toolbox [50]–is based on a pathwise iteration [51]. However, in special cases, more time-consuming algorithms are needed when the number of points in the space of hyperparameters must be kept low. In addition, in this space there is no guarantee of a single minimum, and the local minima can be narrow. Hence, instead of a pre-defined grid, an adaptive algorithm for exploring the details around the minima would be advantageous, as it would also take into account the stochastic nature of MSEs obtained from a CV.

An excellent novel procedure recommended for hyperparameter selection is Bayesian optimization (BO) [41, 42]. Briefly, in this method the function to be optimized is modeled as a sample from a Gaussian process, characterized by a proper kernel (or covariance) function ensuring smoothness. According to the machinery of Bayesian inference [52], the values of $f(\lambda, \omega)$ at the set of known points define a prior probability based on the Gaussian process for the value at any other point. This information is utilized by an acquisition function in determining the position of the next point at which $f(\lambda, \omega)$ is to be calculated to maximize knowledge about the position of its global minimum. Evaluation of $f(\lambda, \omega)$ at the new point results in a posterior probability that in turn will be used to update the prior for selection of the next point. As a result, this algorithm adaptively explores the areas around the potential minima much more extensively than it does in other regions.

## Methods

### Simulation of the absorption kinetics data from a model of the bR photocycle

In the past several decades, numerous photocycle schemes have appeared in the literature for both the wild type and various mutant bRs based on kinetic visible absorption spectroscopy. Single and parallel schemes differ in that parallel schemes assume that the sample is heterogeneous with different bR species going through different photocycles [53], whereas single schemes assume a homogeneous sample [6]. Even single cycles can be branching [3] or non-branching. The complex kinetics of the intermediates have ruled out a single, unidirectional photocycle and the reversibility of most of the molecular steps is now generally accepted [6, 54, 55]. Various strategies have been proposed and applied to find the appropriate photocycle and the corresponding microscopic rate constants [4, 7, 56].

In this study, the simulated data were considered to have been generated by a complex photocycle scheme (Scheme 2), which contains mechanistically necessary steps for the proton pumping function of bR, identified by experiments. This scheme is the result of a synthesis of previously published, linear, reversible schemes with certain additions. The **K** matrix built from the microscopic rate constants of the transitions is listed in S1 Table. The ultrafast transitions involving the "hot" I and J intermediates were not taken into account, as the published multichannel absorption kinetic data generally start in the 10–100 ns range or later, when these intermediates have already decayed to the K intermediate(s) [10]. Two early, sequential L intermediates, L1 and L2, kinetically and spectrally identified as separate intermediates, were considered in accordance with previous work [57]. Relaxation of L1 to L2 is accompanied by the partial reorientation of the Schiff base NH bond, as revealed by X-ray structural data [58]. Since it is generally observed that K persists much longer than the decay of L1, and L1 was shown to completely decay to L2, the model also included a second, spectrally identical K intermediate in equilibrium with L2. A rationale would be the existence of an initial "hot" K, which decays to L1, and both forms can relax to K2 and L2, respectively, probably by energy dissipation into the protein environment of the chromophore. Therefore, these latter steps were considered unidirectional.

**Scheme 2**. **Model of the bR photocycle providing input data for the simulations**. All dark reactions between the intermediates are supposed to be first-order ones. See S1 Table for the corresponding microscopic rate constants.

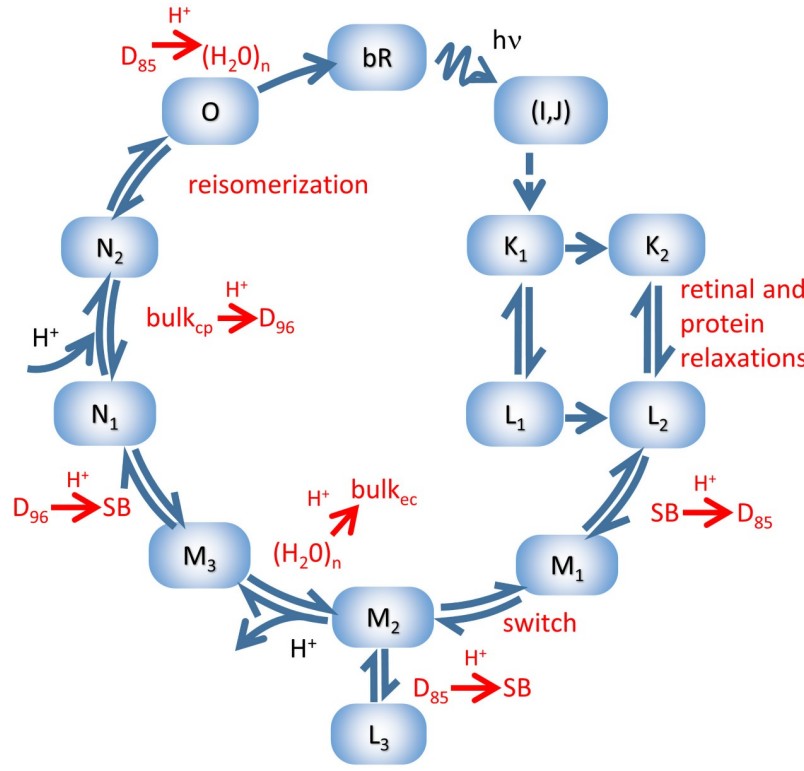

It is assumed that the proton transfer from the Schiff base to the anionic D85 (yielding the M1 intermediate) is followed by the reorientation "switch", resulting in the change of access to the retinal Schiff base from the extracellular to the cytoplasmic side (M2). The model allows proton transfer between D85 and the Schiff base even after the "switch". Therefore, we included in the scheme an L3 intermediate, spectrally unresolved from L2, as a cul-de-sac from M2. The pathway of the proton in this reaction does not need to retrace the original proton transfer from the Schiff base to D85, and it is assumed that the equilibrium shifts towards Schiff base deprotonation (i.e. M2). The model corresponds to neutral or alkaline pH, where proton release from the extracellular water molecule cluster with pKa = 5.8 takes place [59] as a transition between M substates, i.e., no branching to a low pH path with late proton release was modeled. The protonation equilibrium between the Schiff base and D85, i.e., the equilibrium between L and M, is expected to shift completely in favour of Schiff base deprotonation after extracellular proton release, due to the mutual effect of the protonation state of D85 and the proton release cluster on their respective proton affinities [60]. The two sequential N states appear by the reprotonation of the Schiff base by D96 and the proton uptake from the cytoplasmic side to D96. These substates have been experimentally separated at high pH by polarized spectroscopy [61] or in mutants [62]. The substates of M and the substates of N were modeled with a single M and N spectrum, respectively. After N2, the recovery of the initial resting state was considered in the model through a single O intermediate. Experimental evidence shows that at the end of the photocycle, it is difficult to separate M, N and O by visible spectroscopy, a circumstance also complicated by the recovery of the

resting state, so this is a realistic trade-off. Based on infrared spectroscopy, the reprotonation of D96 from the cytoplasmic bulk has been reported to take place in two steps [8]. However, in our model we did not consider the reported splitting of the cytoplasmic bulk to D96 proton transfer into two separate steps. The transfer from N2 to the O intermediate corresponds to retinal reisomerization. The final, unidirectional transition to the resting state combines internal proton transfer from D85 to the extracellular proton release cluster and reversal of any conformational alterations present in the O state [5]. In the simulation we used realistic intermediate spectra extracted from experiments. The spectra obtained earlier by singular value decomposition with self-modeling, cf. Fig 4 in [57] and S2 Fig, were fitted by the analytical nomogram function for visual pigment spectra [63] to obtain noise-free intermediate spectra (S1A Fig). The difference spectra of the intermediates were calculated by subtracting the spectrum of the bR resting state from the intermediate spectra (S1B Fig). The kinetics of the individual intermediates (S1C Fig) were calculated by solving the eigenvalue problem of the **K** matrix as discussed in the Introduction, supposing that at $t = 0$ all intermediate concentrations are zero except for K1. The observable absorption change at time $t$ and wavelength $w$ was calculated as

$$\Delta A(t, w) = \sum_{i=1}^{11} s_i(w)c_i(t) = \sum_{i=1}^{11} D_i(w)e^{-\frac{t}{\tau_i}}, \tag{9}$$

where $s_i(w)$ and $c_i(t)$ are the difference spectrum and the concentration of the $i^{\text{th}}$ intermediate, $\tau_i$ are the macroscopic time constants (S2 Table left column), and $D_i$ is the corresponding DADS. (Formally, an additional, 12$^{\text{th}}$ component of infinite time constant and uniformly zero DADS can be included in the sum on the right side of Eq (9), related to the existence of the bR resting state.) $t$ was sampled at 50 points as 9 logarithmically equidistant points per decade in the domain from 100 ns to 43 ms, while $w$ was sampled at 38 points in the 355–730 nm range.

The effect of random noise in the data on the analysis was tested first by adding uniformly distributed noise of varying amplitude (i.e., standard deviation of its Gaussian distribution relative to the maximal absolute value of the data matrix, $\sigma_{\text{rel}}$) to the data matrix. Finally, realistic noise spectra were constructed by first filling a matrix of the same size as the data matrix, $\Delta A(t, w)$, with Poisson distributed random numbers of mean zero and realistic amplitude. Then, the spectral and temporal variation of the noise amplitude was modeled in a way consistent with our typical experimental conditions. The noise amplitude appearing in the absorption difference spectra is inversely proportional to the square root of the number of accumulated photons, which depends on the light intensity spectrum, on the gate pulse width of the CCD detector and on the number of accumulated scans contributing to a difference spectrum. Hence, the noise spectra (the columns of the matrix) were first multiplied point by point using the inverse of the square root of a typical light intensity spectrum of a tungsten lamp, measured behind a sample of wild-type bacteriorhodopsin (S2A Fig). Usual measurements over many decades in time were split into a few time segments with increasing gate pulse width and a varying number of accumulations, to produce a better signal-to-noise ratio where long delay times allow longer integration times. Accordingly, the simulated noise spectra were then divided by the square root of 5, 15 and 40 in the delay segments of [1 μs, 20 μs], [20 μs, 100 μs], ≥100 μs, respectively, corresponding to typical noise reduction at later delays as compared to the fastest, <1 μs delay range (see S2B Fig). The resulting spectrally and temporally heterogeneous noise matrix was added to the noise-free spectrotemporal data matrix to yield the input data for the analysis with realistic noise.

## Ultrafast fluorescence kinetics measurements on FAD

For comparison with the results presented in our previous study [28], the experimental conditions were kept identical to those described therein. The fluorescence kinetics experiments were carried out on samples of 1.5 mM aqueous solution of FAD disodium salt hydrate (Sigma-Aldrich) in 10 mM HEPES buffer at pH = 7.0. A home-made measuring apparatus combined the techniques of fluorescence up-conversion and time-correlated single photon counting (TCSPC) for the detection of fast and slow components, respectively. The sample was excited at 400 nm with 150 fs pulses of 80 MHz repetition rate. The fluorescence kinetics were detected at magic angle conditions at 11 wavelengths in the 490–590 nm range. The up-conversion technique sampled the kinetics in a linear section of 0.1–1.2 ps with a dwell time of 0.1 ps, followed by logarithmically equidistant section up to 300 ps with a logarithmic dwell time—defined as $\log_{10}(t_{i+1}/1ps) - \log_{10}(t_i/1ps)$–of 0.1. The TCSPC technique sampled in the 0–6.38 ns range with a dwell time of 4 ps, the obtained data were then compressed by averaging into a logarithmically equidistant scale with a logarithmic dwell time of 0.05. The two datasets were merged by fitting a small overlapping section at around 150 ps, resulting in a final one consisting of 69 points and ranging from 0.1 ps to 8.91 ns.

## Simulation of distributed kinetics data

This simulation was based on hypothetical reaction kinetics following the Arrhenius equation

$$k = Ae^{-\frac{E}{RT}}. \tag{10}$$

In arbitrary units, Eq (10) can be expressed as

$$k(E) = e^{-\frac{E}{50}}, \tag{11}$$

where the activation energy $E$ was sampled in the interval [0,400] with increment 1 (S3A Fig red line). It was supposed that the molecular population cannot be characterized by a discrete value of the activation energy, but—due to the existence of an assembly of substates—it can be described by a Gaussian distribution $g(E)$ with a mean of 200 and $\sigma = 35$ (S3A Fig blue line). The simulated true solution (the distribution of $g(E)$ over $\tau = 1/k(E)$) is presented in S3B Fig.

## Implementation of the machine-learning procedure

An object-oriented MATLAB toolbox FOkin (First-Order kinetics) was developed to handle all the simulation, parameter estimation and model selection problems in a common software environment. The detailed description of the toolbox is included in its documentation. Here we outline the main procedures applied therein.

A fundamental task in our simulation is solving the group lasso problem ($\omega = 1 - \alpha = 0$) or the GENP ($0 < \omega \leq 1$) defined in Eq (8). To this end, we tested the following algorithms:

- the blockwise descent algorithm [64] implemented in the glmnet MATLAB package [50];

- the simultaneous signal decomposition formulation based on block-coordinate descent implemented in the SPAMS toolbox [65];

- the fast iterative shrinkage-thresholding algorithm (FISTA) [66] implemented in the SPAMS toolbox;

- the alternating direction method of multipliers (ADMM) [31] algorithm implemented in a collection of MATLAB functions [67].

Surprisingly, the first three algorithms, which perform well for other problems, showed slow convergence, and/or optimized with poor sparsity on our design matrix. On the other hand, the *group_lasso* function [67] implementing the ADMM algorithm provided excellent sparsity with a reasonable convergence rate. The convergence rate improved further when we incorporated simple criteria for the adaptive updating of the augmented Lagrangian penalty parameter [31]. The augmented Lagrangian technique applied in ADMM also offered a trivial way for the inclusion of the squared $L_2$ penalty term in Eq (8). With these modifications, the function was incorporated into the FOkin toolbox, and all GENPs in this study were solved by that. These calculations were carried out with a time constant vector $\boldsymbol{\tau}$ of 50 logarithmically equidistant points in a decade. If the solution of a GENP was sparse, it was discretized by characterizing every contiguous region of nonzero values by a time constant and an amplitude, independently at each wavelength. The time constant was determined by averaging the elements of the $\boldsymbol{\tau}$ vector falling into the region, applying the absolute value of the corresponding amplitudes as weighting factors. The amplitude was calculated by adding the (signed) amplitudes of the contributing individual elements.

With a dataset taken at a single wavelength, the group lasso penalty in Eq (8) turns into the simple lasso formula. In this case, the minimization problem can be solved by the Primal-Dual interior method for Convex Objectives (PDCO) [29], implemented in the PDCO MATLAB function [68]. (An earlier version of PDCO was incorporated in the SparseLab toolbox [69] and utilized in our previous study [28].) Since PDCO outperforms ADMM in runtime, this algorithm was also incorporated into FOkin, and in this study it was applied for the analysis of distributed kinetics.

Both the $k$-fold CV and the RCV($n_v$) algorithms were implemented in FOkin from scratch, applying parallel calculation on the available CPU cores. The random sorting of the data points into the training and testing groups was carried out independently at every wavelength. In $k$-fold CV the value of $k$ was 10, as mostly used in the literature. In RCV($n_v$) the fraction $n_c$ / $n$ was 0.9, and the MSE was calculated by averaging the results of $10^4$ CVs in the restricted space.

The machine-learning procedure was based on the automatic hyperparameter selection by BO using the *bayesopt* function of the Statistics and Machine Learning Toolbox of MATLAB, which applies the ARD Matérn 5/2 kernel [70]. The optimal values of both $\lambda$ and $\omega$ were searched on a logarithmic scale. To cover a high dynamic range, the objective function was also transformed to be logarithmic. On the joint space of $\lambda$ and $\omega$, the search was carried out on 400 points. If any of the hyperparameters was fixed, the number of search points typically was 100. The *expected-improvement-plus* type of the acquisition function was applied to reduce the chance of missing the true minimum.

For the analysis of the experimental fluorescence kinetics data, the temporal instrument response function of the measuring device was described by a Gaussian with mean $t_0$ and standard deviation $\sigma$. Accordingly, the pure exponential terms in the columns of the design matrix in Eq (3) were substituted for the analytical function of their convolution with a Gaussian [18]. We supposed that $\sigma$ is wavelength independent and the wavelength dependence of $t_0$ –related mainly to light dispersion—was modeled by a cubic spline of three knot points with fixed x coordinates. The y coordinates of the knots, as well as the value of $\sigma$ were considered as free parameters to be determined from the experimental data. To that end, we set a GENP with the data, a reasonable value of $\lambda$, $\omega = 0$ and a wavelength-dependent series of the design matrices with these parameters, and applied again the BO method to determine their optimal values. For the main machine-learning procedure the obtained values were kept fixed.

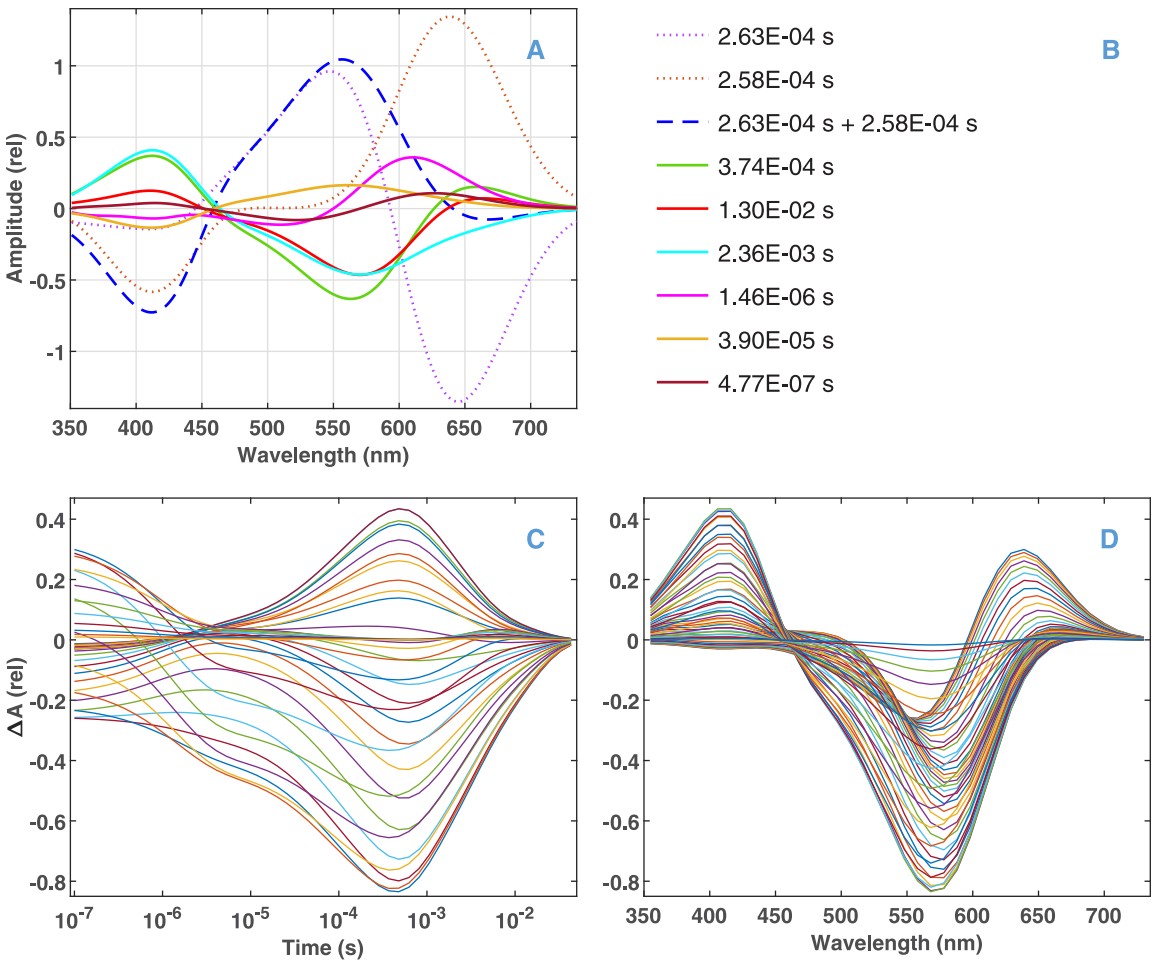

**Fig 1. Input parameters of the simulation based on Scheme 2.** (A) The dominating DADSs and (B) the corresponding macroscopic time constants. (C) Temporal and (D) spectral representation of the kinetic data calculated from the parameters presented in (A) and (B).

## Results and discussion

### Recovering the macroscopic kinetic parameters of a bR photocycle model from simulated data

The data simulated from the bR photocycle model in Scheme 2 pose several challenges for the recovery of the true macroscopic rate constants and DADSs by a machine-learning procedure. First, the amplitude corresponding to three of the true time constants (S2 Table left block in normal face) is less than 3% of the maximal one. (A fourth, very small component with infinite time constant is a calculation error, since it should be of zero amplitude as explained in the description of Eq (9).) The DADSs of remaining dominating components and the corresponding time constants are presented in Fig 1A and 1B, respectively. The second challenge is that the time constant of the two largest DADSs ($2.63 \times 10^{-4}$ s and $2.58 \times 10^{-4}$ s) are obviously unresolvable. In addition, the DADS of these components (dotted lines) show definite mirror symmetry in the high wavelength region, and even their sum (dashed line) remains the largest spectrum. The kinetic data generated from the true parameters are presented in Fig 1C and 1D in temporal and spectral representation, respectively.

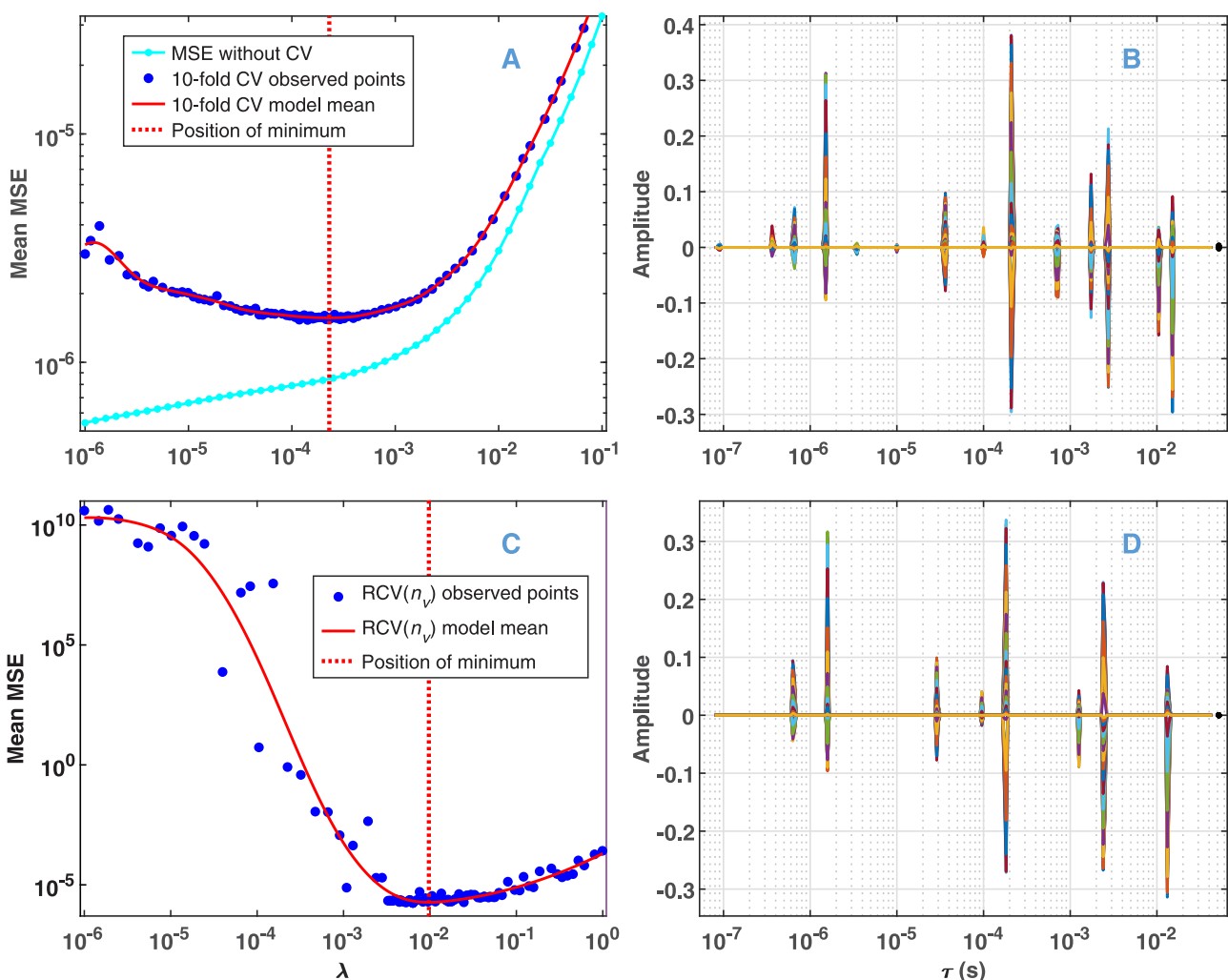

**Fig 2. Model selection by BO from the simulated data presented in Fig 1C and 1D at a fixed value of $\omega = 1$.** Noise level: $\sigma_{rel} = 10^{-3}$. (A) Mean MSE obtained without (cyan) and with 10-fold CV (blue and red). (C) Mean MSE obtained with RCV($n_v$). (B) and (D) Solution of the GENP calculated with the value of $\lambda$ selected at the minimum presented in (A) and (C), respectively. A line in a particular color in panel (B) and (D) corresponds to a column $\mathbf{x}_{\cdot,k}$ in Eq (8), plotted against the grid of the time constants, while the colors represent different wavelengths. (The black dot beyond the upper limit of $\tau$ corresponds to the value of $\tau = \infty$.) The sparsity of the solution is manifested in the low number of features in the form of narrow spikes, representing single exponentials, whose time constant and amplitude are indicated by their location and height, respectively.

The 10-fold CV curve calculated from the above data with a relative noise of $10^{-3}$ is presented in Fig 2A. The blue dots represent the points sampled by BO based on GENP with $\omega = 1$, determining a model mean (points predicted on a dense grid after the sampling process, red curve) which has a well-defined minimum in the space of $\lambda$. In contrast to that, the MSE calculated from the solution of GENP without CV on a grid of $\lambda$ is obviously monotonic (cyan curve). As expected with $\omega = 1$, and in accordance with our previous results [28], the solution corresponding to the minimum is sparse (Fig 2B): it consists of narrow peaks, referred to as features in the following. Also, as a consequence of the grouping penalty, the position of the features is equal at all wavelengths. However, as anticipated, the 10-fold CV is biased towards the more complex models [47]. This manifests in the high number of features and especially in the false doublets at $\sim 2 \times 10^{-3}$ s and $\sim 1 \times 10^{-2}$ s. To eliminate this problem, the model selection procedure was repeated with RCV($n_v$). As seen in Fig 2C, in this case the value of the

selected $\lambda$ is by more than an order of magnitude higher than that preferred by 10-fold CV. Accordingly, the selected model is much simpler (Fig 2D). This finding excellently agrees with the theoretical results of Feng and Yu, as well as with their calculations with simulated and real data [48].

Utilizing the superior performance of RCV($n_v$) we set up the following machine-learning algorithm.

The 3-step BO is needed because BO is time-consuming, and the 2D version with a reasonable number of iterations yields only a rough estimation of the hyperparameters. For simplicity, we refer to these steps as model selection and to Steps 4 and 5 as parameter estimation. Note, however, that the algorithm can also be considered as a hierarchical model selection setting. On one level the hyperparameters are selected by BO, while on the other the nonzero elements are selected from the solution of the GENP.

Algorithm 1 was used to analyze the simulated data with relative noise ranging from $10^{-7}$ to $10^{-2}$. The detailed results are summarized in S2 Table, here we compare the characteristics obtained with a small ($10^{-7}$) and large ($10^{-3}$) value of $\sigma_{rel}$. The results of the 3-step model selection are presented in Fig 3, which also indicates the model and noise errors of BO. Since $\lambda$ controls mainly the number of features, while $\omega$ their width, and consequently the support size averaged over the wavelength space, these parameters are also shown (purple curves). It can be clearly seen that the algorithm automatically handles the presence of noise in an optimal way. A higher noise level would cause unjustified complexity in the solution of the GENP, but the selected hyperparameters—much higher $\lambda$ and somewhat lower $\omega$–effectively compensate for this unwanted tendency. In addition, the selected value of $\omega$ in both cases is very low: it falls into the range where the average support size is around its minimum, ensuring perfect sparsity.

**Algorithm 1**

```
1. Execute a 2D BO optimization on a wide range of both λ and ω, apply-
ing RCV(nᵥ) with the GENP.
2. Fix the optimal value of λ obtained in Step 1 and execute a BO on ω
only.
3. Fix the optimal value of ω obtained in Step 2 and execute a BO on λ
only.
4. With the optimal value of ω and λ obtained in Steps 2 and 3, solve
the GENP.
5. If the solution in Step 4 is sparse, discretize it.
```

The results of Steps 4 and 5 of Algorithm 1 are shown in Table 1 and Fig 4A and 4B, and are set against the true values. For a compact presentation, the amplitudes in Table 1 represent the maximum absolute value of the corresponding DADSs shown in Fig 4B. According to the table, at $\sigma_{rel} = 10^{-7}$ all the ten finite and resolvable true components are recovered by the algorithm. In addition, the algorithm yielded six false positive features with low amplitudes. Neglecting the components with amplitudes smaller than 5% of the largest one, seven true and one false positive features remain (bold in the table and plotted in Fig 4B). In this selection, the remaining false positive feature is the lowest one (black in Fig 4B). As indicated by red arrows in Fig 4A, the position of all valid features is very close to that of their true counterparts. At $\sigma_{rel} = 10^{-3}$ 7 finite and resolvable true components are recovered with one false positive component. The above neglect affects one true feature. In this case the position of the recovered features is less perfect, and the one at $3.74 \times 10^{-4}$ s is considerably shifted.

The DADSs derived in Step 4 (Fig 4B) perfectly recover the shape of the true ones. However, the amplitude of the two largest spectra (corresponding to the true components of $2.61 \times 10^{-4}$ s and $3.74 \times 10^{-4}$ s) is considerably smaller than that of their true counterparts, especially in the case of high noise. Obviously, this anomaly is an inherent drawback of

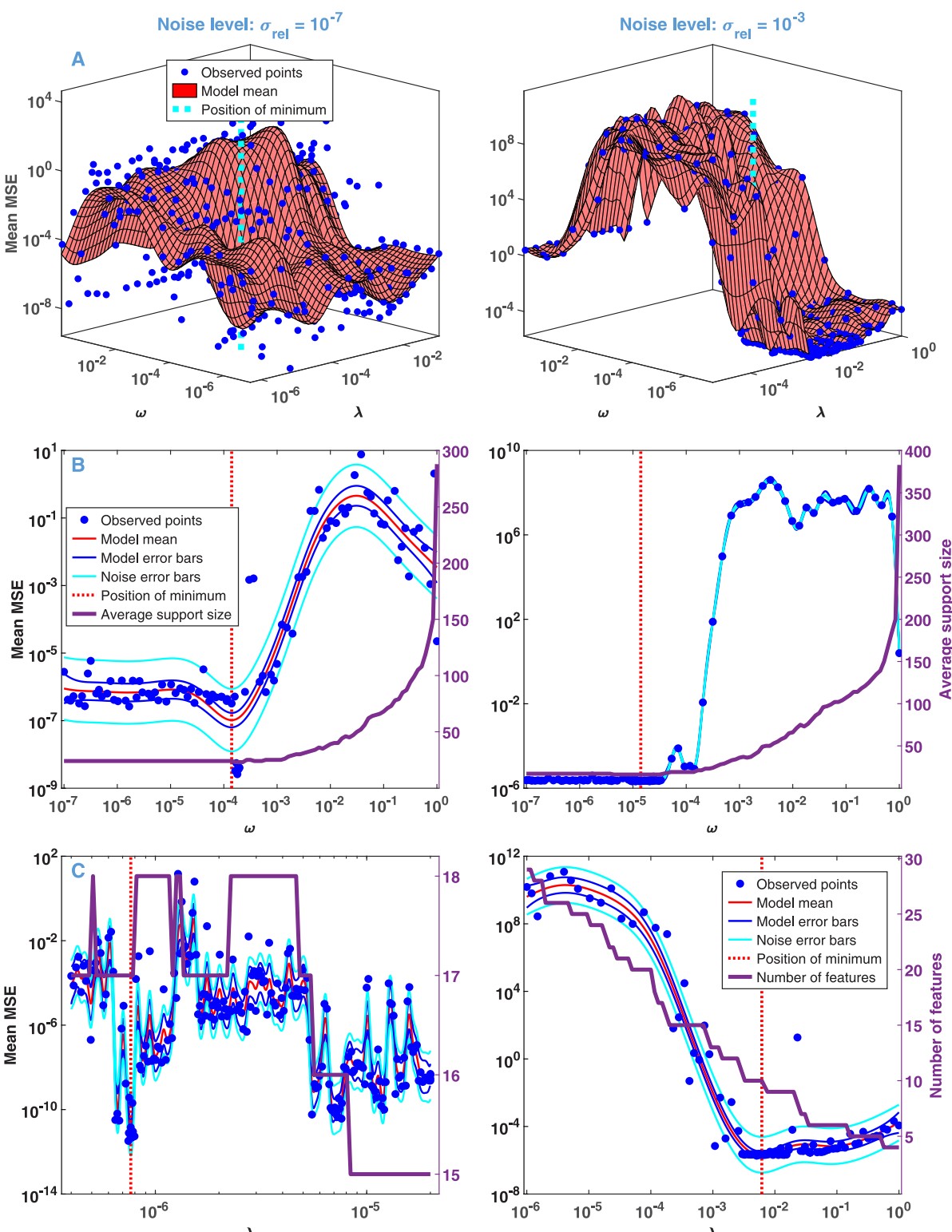

**Fig 3. Model selection by BO based on RCV(nv) from the simulated data with different noise levels.** Results of the steps of Algorithm 1. (A) Step 1: BO in the joint space of $\lambda$ and $\omega$, explored by the BO process at the blue dots. (B) Step 2: BO in the space of $\omega$ with the fixed value of $\lambda$ obtained in Step 1. The purple line (right axis) represents the average support size against $\omega$. (C) Step 3: BO in the space of $\lambda$ with the fixed value of $\omega$ obtained in Step 2. The purple line (right axis) represents the number of features against $\lambda$.

**Table 1. Kinetic parameters (τ and A) predicted at low and high noise levels on the simulated data by Algorithm 1 and Algorithm 2 (8 exponentials).** MSE refers to the mean square error of the fit.

| true values | | $\sigma_{rel}$ = 1.E-7 | | | | $\sigma_{rel}$ = 1.E-3 | | | |
|---|---|---|---|---|---|---|---|---|---|
| | | Algorithm 1 | | Algorithm 2 | | Algorithm 1 | | Algorithm 2 | |
| | | MSE = 1.146E-10 | | MSE = 1.134E-08* | | MSE = 1.497E-06 | | MSE = 6.744E-07 | |
| τ (s) | A | τ (s) | A | τ (s) | A | τ (s) | A | τ (s) | A |
| 1.67E-07 | 3.E-05 | 1.94E-07 | 0.003 | | | | | | |
| 3.37E-07 | 0.028 | 3.29E-07 | 0.020 | | | | | | |
| **4.77E-07** | **0.107** | **5.19E-07** | **0.095** | **4.97E-07** | **0.115** | **6.23E-07** | **0.153** | **6.23E-07** | **0.160** |
| **1.46E-06** | **0.359** | **1.48E-06** | **0.357** | **1.53E-06** | **0.367** | **1.57E-06** | **0.324** | **1.62E-06** | **0.322** |
| 2.65E-06 | 0.026 | 2.77E-06 | 0.022 | **1.99E-06** | **0.065** | 3.98E-06 | 0.001 | | |
| **3.90E-05** | **0.164** | **3.90E-05** | **0.157** | **3.92E-05** | **0.163** | **3.02E-05** | **0.120** | **3.83E-05** | **0.162** |
| **2.61E-04**** | **1.045** | **2.34E-04** | **0.737** | **2.53E-04** | **0.856** | **1.85E-04** | **0.507** | **2.56E-04** | **0.839** |
| **3.74E-04** | **0.632** | **4.68E-04** | **0.325** | **4.07E-04** | **0.451** | **1.15E-03** | **0.104** | **4.15E-04** | **0.448** |
| **2.36E-03** | **0.462** | **2.35E-03** | **0.468** | **2.38E-03** | **0.458** | **2.41E-03** | **0.435** | **2.38E-03** | **0.454** |
| **1.30E-02** | **0.464** | **1.30E-02** | **0.463** | **1.31E-02** | **0.461** | **1.32E-02** | **0.457** | **1.30E-02** | **0.463** |
| Inf | 1.E-15 | Inf | 3.E-05 | | | Inf | 4.E-04 | | |
| | | **False positive components** | | | | | | | |
| | | 7.94E-08 | 0.001 | | | | | | |
| | | 1.44E-05 | 0.001 | | | | | | |
| | | 5.08E-05 | 0.018 | | | | | | |
| | | **1.16E-04** | **0.040** | | | **1.00E-04** | **0.044** | **1.43E-04** | **0.058** |
| | | 9.55E-04 | 0.034 | | | | | | |
| | | 3.31E-03 | 0.002 | | | | | | |

*MSE of the corrected fit keeping all the 17 components is 6.134E-11

**unresolved components of 2.58E-04 s and 2.63E-04 s

components in bold refer to those kept after discretization

Algorithm 1, based on the concept of fitting with penalties. Namely, by definition both the $L_1$ and $L_2$ penalties have a higher controlling effect on the components of higher amplitudes. The only way to correct this unwanted side effect is to lift the constraint imposed by them and execute a simple exponential fitting, the method we highly argued against in the Introduction. Note, however, that at the present point of the analysis this step is completely justified on the grounds of information already acquired by Algorithm 1. In fact, as the solutions are sparse, we can confirm that the first-order model is correct. We also know the number of the exponentials and even their parameters with great accuracy. This means that the solution obtained by Algorithm 1 is very close to the global minimum of the pure multiexponential fitting problem. Accordingly, we suggest the following extension of the algorithm.

The results of Algorithm 2 with the eight dominating components kept above are presented in Table 1, by blue arrows in Fig 4A and 4C. In the case of $\sigma_{rel} = 10^{-7}$, the positions of the components were almost correct already after Step 1 and they hardly changed in Step 2. The false positive component disappeared, while a neglected one was recovered. At $\sigma_{rel} = 10^{-3}$, the anomalous shift from $3.74 \times 10^{-4}$ is perfectly compensated. In both cases, the diminished DADS amplitudes become closer to the true values. Notably, the solutions at the two noise levels become very similar. The detailed results at a single wavelength are shown in S4 Fig. Keeping all the 17 components presented in Table 1 under $\sigma_{rel} = 10^{-7}$ and executing the exponential fitting with them causes minimal changes in the DADSs of the dominating ones (S5 Fig). Compared to the result of Step 1, after Step 2 the MSE reduced by a factor of two. Overall, for

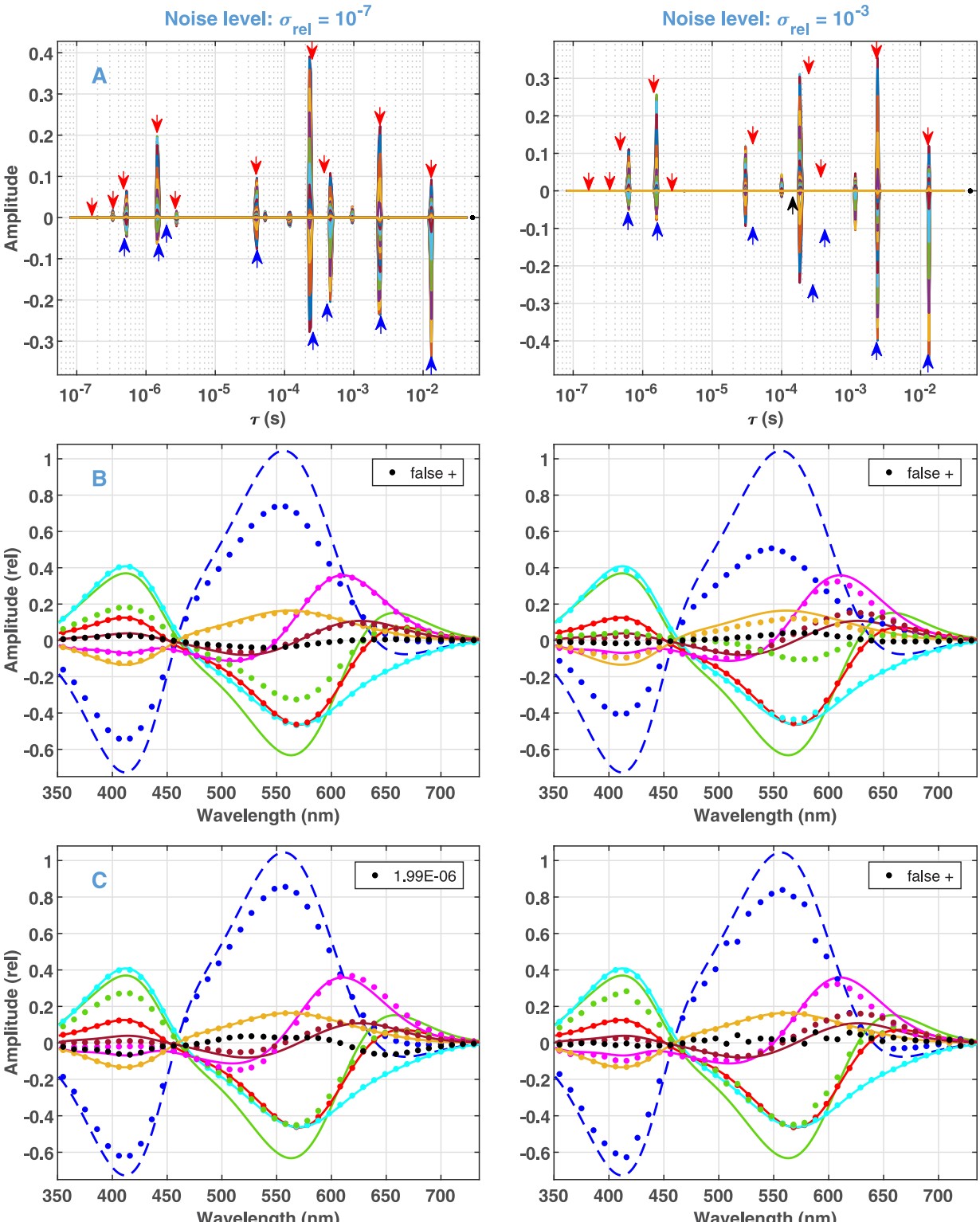

**Fig 4. Results of Algorithms 1 and 2 on the selected models presented in Fig 3.** (A) Representation over time constants (Algorithm 1, see captions for Fig 2B and 2D for details). The arrows point to the position of the true values (red) and those obtained by Algorithm 2 (valid ones blue, false positive black). (B) Representation by DADSs, neglecting components below 5% of the maximal one (Algorithm 1, dotted) compared to the true DADSs as presented in Fig 1A. (C) DADS obtained by Algorithm 2.

the dominating components, the correction by exponential fitting brings the estimated parameters considerably closer to the true values and reduces the difference in the solution at different noise levels.

**Algorithm 2**

```
1. Execute Algorithm 1.
2. Execute a global multiexponential fitting with the result of dis-
cretization obtained in Step 1 as starting parameters.
```

Up to this point it was assumed that noise has the property of independent and identical distribution (*iid*) for the whole spectrotemporal dataset. However, this criterion may not be satisfied in real experimental data for two main reasons:

1. The number of molecules participating in a chemical reaction network can be very low (e.g., in a single cell). In this case, the occupancies of the species become partially stochastic [71]. This noise-like component clearly depends on both time and species.

2. The experimental procedure introduces noise depending on time and/or wavelength.

For the absorption kinetics of the bR photocycle, the observation of the first effect is completely unrealistic with the present experimental techniques. According to the method of linear noise approximation, the ratio of the classical deterministic and the stochastic part of the kinetics is of order $\sqrt{N}$, where $N$ is the number of participating molecules [71]. To obtain a reasonable absorption kinetics signal of the bR photocycle, one needs to excite ~$10^{14}$ bR molecules; hence the estimated relative level of noise originating from this intrinsic effect is $\sigma_{rel} = 10^{-7}$. The dominating stochastic error in a real absorption kinetics experiment is due to the photon noise of the observation beam. As described in the Methods and shown in S2 Fig, the level of this noise depends markedly on both time and wavelength. In a real experiment, the average of this noise is usually of order $10^{-3}$–$10^{-2}$ relative to the maximal amplitude of the measured signal [3, 57, 59]. The time constants and amplitudes predicted for the data simulated with average noise at $\sigma_{rel} = 3.5 * 10^{-3}$ are presented in S3 Table. Comparison with S2 Table shows that these parameters fall very well between those corresponding to the cases with *iid* noise of $\sigma_{rel} = 10^{-3}$ and $\sigma_{rel} = 10^{-2}$, proving that the algorithm works perfectly even when the *iid* condition is not satisfied.

In summary, it was found that Algorithm 2 is a very sophisticated method to recover the macroscopic time constants and DADSs corresponding to a very complicated photocycle model in a wide range of error levels, provided that the absolute value of their amplitudes reaches at least a few percent of the maximal amplitude. Under this low threshold level, both positive and negative false components emerge. The selected model of the bR photocycle leads to three such minor components; hence their justification from the experimental data seems to be unsolvable even at a very low noise level. The analysis of such real experimental data by the methods applied in this study will be published elsewhere.

## Analysis of ultrafast experimental fluorescence kinetic data on FAD

The input dataset obtained by ultrafast fluorescence kinetic measurements is presented in Fig 5. On the strength of the knowledge base gathered on the above simulated data, we applied Algorithm 2 to this experimental dataset. As seen in Fig 6, Steps 1 to 3 of Algorithm 1 select a low value for $\omega$ (leading to a minimal support size), and a high value of $\lambda$ (corresponding to five features). The final results of Algorithm 2 are presented in Table 2 and Fig 7, with details in S6 Fig. As expected from the hyperparameters, the solution after Step 1 of Algorithm 2 is very sparse (Fig 7A). All finite time constants are kept after discretization. Both the time constants and the DADSs (Table 2, Fig 7B) are similar to what was presented in Figs 7 and 8 of our

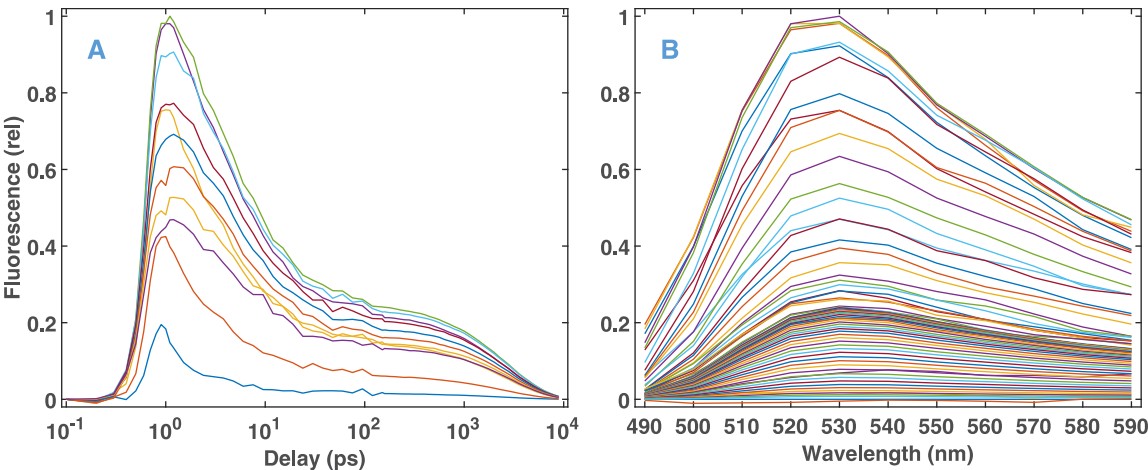

**Fig 5. Experimental fluorescence kinetic data on FAD.** (A)Temporal and (B) spectral representation.

previous study [28] carried out on a dataset obtained from a similar experiment and analyzed by lasso with a manually selected value of $\lambda$. The main difference is the complete wavelength independence of the time constants due to the group lasso penalty applied in the present study. For the same reason, here the shape of the DADSs is smoother. Step 2 of Algorithm 2 hardly affected the position of the features (arrows in Fig 7A), but considerably increased the DADS corresponding to time constants of 8.4 ps and 0.35 ps. Meanwhile, the MSE changed by less than a factor of two (Table 2, S6C and S6D Fig). Overall, Algorithm 2 performed similarly on the experimental FAD fluorescence data and on simulated data, in spite of the fact that the latter corresponded to an entirely different kinetic process modeling the bR photocycle. The comparison of the MSE values presented in Table 2 and S2 Table indicates—as seen in S6 Fig too—that the noise level on the experimental data is within the range used in the simulations. The drop in the level of the residuals above 100 ps (S6C and S6D Fig) is due to the change of the measuring technique from fluorescence up-conversion to TCSPC.

## Analysis of distributed kinetics

The excellent model selection properties of Algorithm 2 on the above sparse kinetics naturally raise the question: how does it behave if the underlying distribution is not actually sparse? Indeed, conformational heterogeneity in proteins can be manifested in distributed kinetics in their functions like ligand binding [2] or folding [72]. Following such kinetics, the truly exciting question is whether Steps 1 to 3 of Algorithm 1 will force on them a relatively good approximation with a sparse distribution or they will be able to automatically adjust the value of $\omega$ high enough to yield the correct dense solution. To answer this question, we simulated a hypothetical dense distribution over the time constants as described in Methods and depicted in S3B Fig. As shown in Fig 8, the corresponding kinetic curve is hardly distinguishable from that calculated from a single discrete value at the maximum of the simulated activation energy distribution (S3A Fig blue line). The analysis of the simulated distributed kinetics was carried out with relative noise levels of $10^{-4}$, $10^{-3}$ and $10^{-2}$.

The results of Steps 1 to 3 of Algorithm 1 with $\sigma_{rel} = 10^{-3}$ are presented in the left column of Fig 9. The algorithm behaved as it did for the truly sparse distributions by selecting a very low value of $\omega$ in the range of the minimal support size. Accordingly, the solution obtained in Step 4 consists of 3 features of minimal width in the range of the true dense distribution. The residual of the fit shows an anomalously uneven structure (Fig 10 left column).

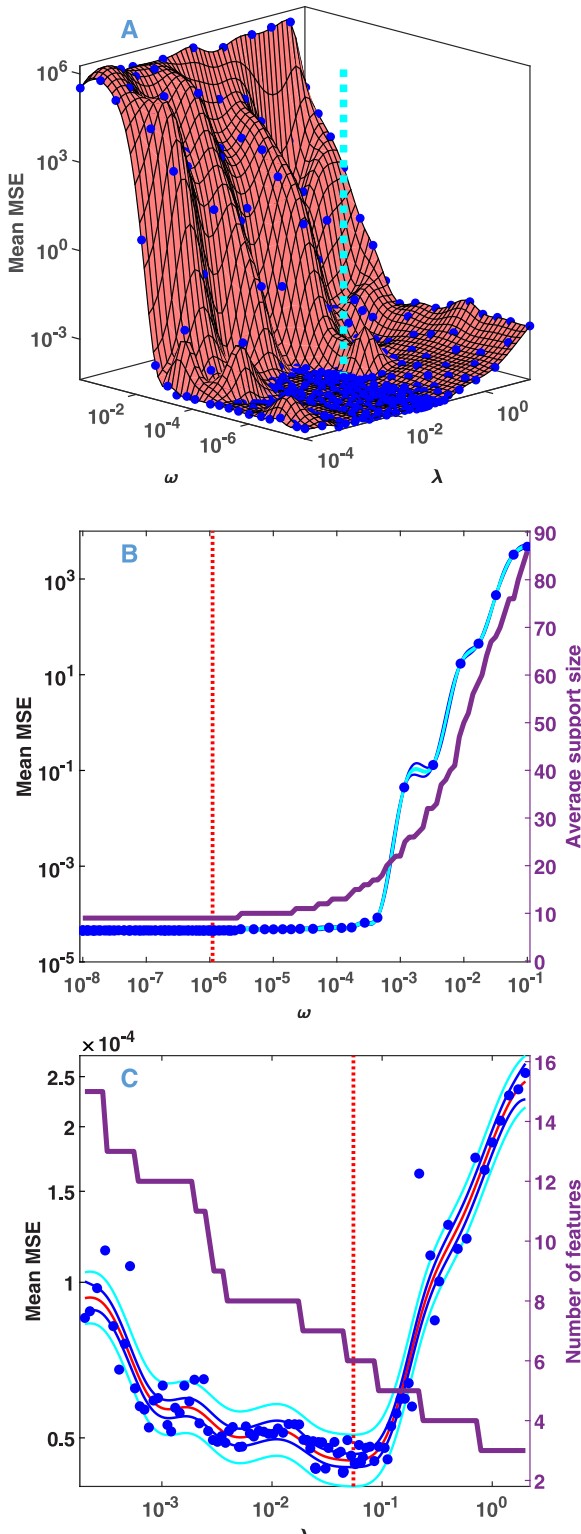

**Fig 6. Model selection from the fluorescence kinetic data presented in Fig 5.** For details see the legends and caption of Fig 3.

**Table 2. Kinetic parameters (τ and A) predicted from the fluorescence kinetic data by Algorithm 1 and Algorithm 2 (5 exponentials).**

|  | Algorithm 1 | | Algorithm 2 | |
|---|---|---|---|---|
| $\lambda$ | 5.5E-02 | | | |
| $\omega$ | 1.1E-06 | | | |
| MSE | 5.044E-05 | | 2.978E-05 | |
|  | **τ (ps)** | **A** | **τ (ps)** | **A** |
|  | **4.5E-01** | **0.171** | **3.5E-01** | **0.213** |
|  | **3.2E+00** | **0.436** | **2.0E+00** | **0.393** |
|  | **1.0E+01** | **0.301** | **8.4E+00** | **0.408** |
|  | **5.4E+01** | **0.077** | **5.1E+01** | **0.085** |
|  | **2.8E+03** | **0.253** | **2.8E+03** | **0.253** |
|  | Inf | 0.001 | | |

Components in bold refer to those kept in discretization

As a matter of fact, the above failure of Algorithm 1 is not surprising. It is based on the RCV($n_v$) procedure, which was preferred over 10-fold CV just because it selects simpler models. Apparently, our purpose now is to move to the opposite direction towards the more 'complex' solutions. To explore this route, the model selection steps were repeated with 10-fold CV instead of RCV($n_v$), the results of which are presented in the right column of Fig 9. According to panel B, this method selected $\omega = 1$, involving all points of the **τ** vector into the support. At the same time, the selected $\lambda$ moved down compared to that selected by RCV($n_v$) (Fig 9C). The solution of the GENP calculated with these hyperparameters results in a single broad feature, beyond a negligible one at $\tau = \infty$. The corresponding residual is random with the expected error level (Fig 10 right column). The obtained distribution is practically indistinguishable from the true one up to $\sigma_{rel} = 10^{-3}$ and kept reasonably similar even at $\sigma_{rel} = 10^{-2}$ (Fig 11). In contrast to these results on the dense true distribution, retesting the bR and FAD data analyzed above with 10-fold CV still resulted in low values of $\omega$, similar to those obtained by RCV($n_v$), hence not jeopardizing the conclusions drawn above.

Note that the dense character of the obtained solutions does not mean that the underlying model is more complex than a sparse one, e.g., the one shown in the left panel of Fig 10A. The solution is dense only in the sense that it is represented by a wide support in the space of our pre-selected points of time constants. However, this is not a fortunate representation of a Gaussian (the true distribution) that can be characterized by only three parameters (location, width, amplitude) in the continuous space, hence satisfying the principle of parsimony. In contrast to that, the sparse solution after discretization needs six parameters (three locations and three amplitudes) for its characterization.

In summary, the model selection based on 10-fold CV recovered a dense true distribution as correctly as RCV($n_v$) did with a sparse one. However, a dense solution means that our fundamental hypothesis that the true model is sparse failed; hence it is better to look for other types of models that can be characterized by a low number of parameters. On the grounds of the results of this study, the suggested final algorithm for the analysis of unknown kinetics hypothesized to be of first order is Algorithm 3.

### Algorithm 3

```
1. Execute a 2D BO optimization on a wide range of both λ and ω, apply-
ing 10-fold CV with the GENP.
2. Fix the optimal value of λ obtained in Step 1 and execute a BO on ω
only.
```

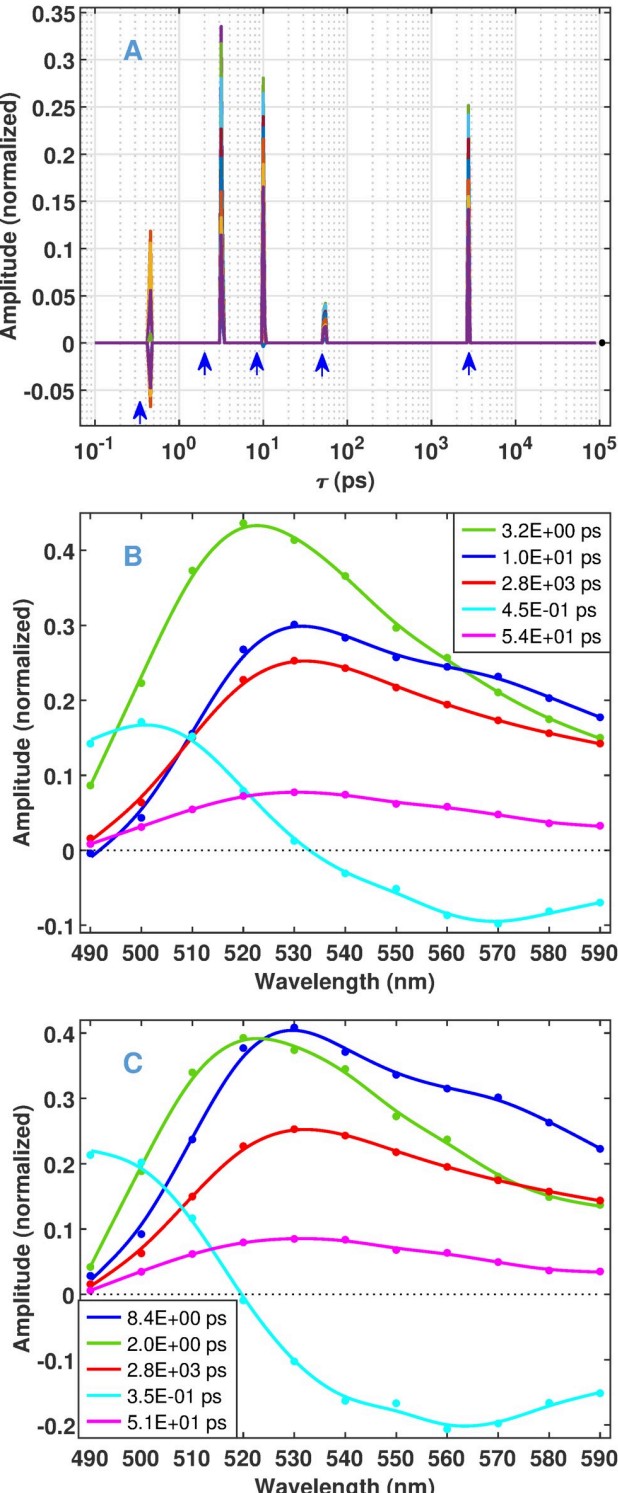

**Fig 7. Fluorescence kinetic parameters predicted by the selected models presented in Fig 6.** For details see the caption of Fig 4. In (B) and (C) the continuous lines are smoothing splines over the data plotted by dots.

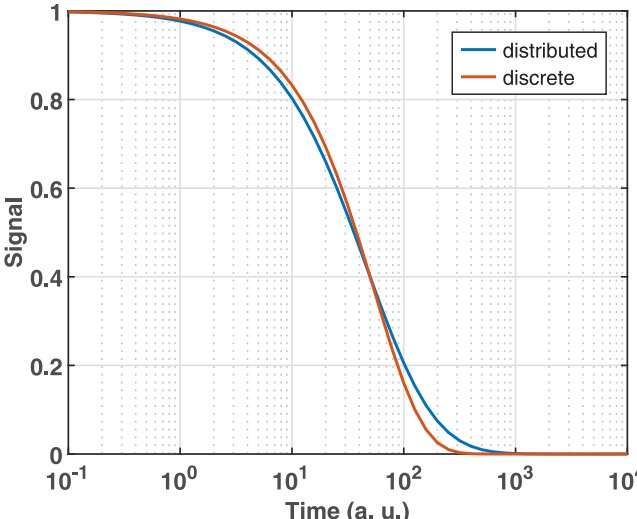

**Fig 8. The kinetics corresponding to the distributions presented in S2 Fig (blue).** For comparison see the kinetics corresponding to the single discrete value at the maximum (200) of the activation energy distribution (red).

```
3. If the value of ω is not low, the expected solution is not sparse.
Go to Step 6.
4. The expected solution is sparse. Execute Algorithm 2.
5. Return.
6. The algorithm reached the limits of its valid range: the kinetics
cannot be described by first-order reactions. It is still worthwhile
to execute Steps 1 to 4 of Algorithm 1. Depending on the shape of the
solution obtained, try to analyze the data with different sorts of
models.
```

It is beyond the scope of this study to investigate how low the value of $\omega$ should generally be in Step 3 of this algorithm to ensure sparsity. As a guideline, Figs 3B and 6B indicate that $\omega = 10^{-4}$ is a safe upper limit.

## Possible extensions

Our results can be extended in many directions. The most obvious extensions are cases when an eigenvalue of rate matrix **K** is degenerate with multiplicity of $n$, leading to a term of

$$At^{n-1}e^{-kt}. \tag{12}$$

appearing in the solution of Eq (1). In the special case of $A = k^n / (n-1)!$ formula (12) is called Erlang distribution, which—independently of first-order reactions—can be used e.g. for modeling intracellular processes with molecular memory [73]. The simplest first-order reaction leading to formula (12) with $n = 2$ requires three species in the scheme

$$S_1 \xrightarrow{k} S_2 \xrightarrow{k} S_3. \tag{13}$$

To test Algorithm 3 for such a term, alone and mixed with true exponentials, we simulated data in the form

$$te^{-\frac{t}{\tau_1}} + A\left(e^{-\frac{t}{\tau_2}} + e^{-\frac{t}{\tau_3}}\right). \tag{14}$$

with both zero and nonzero value of $A$ in the presence of low and high noise levels. The solutions of the GENP with hyperparameters selected by $RCV(n_v)$ and 10-fold CV are presented in

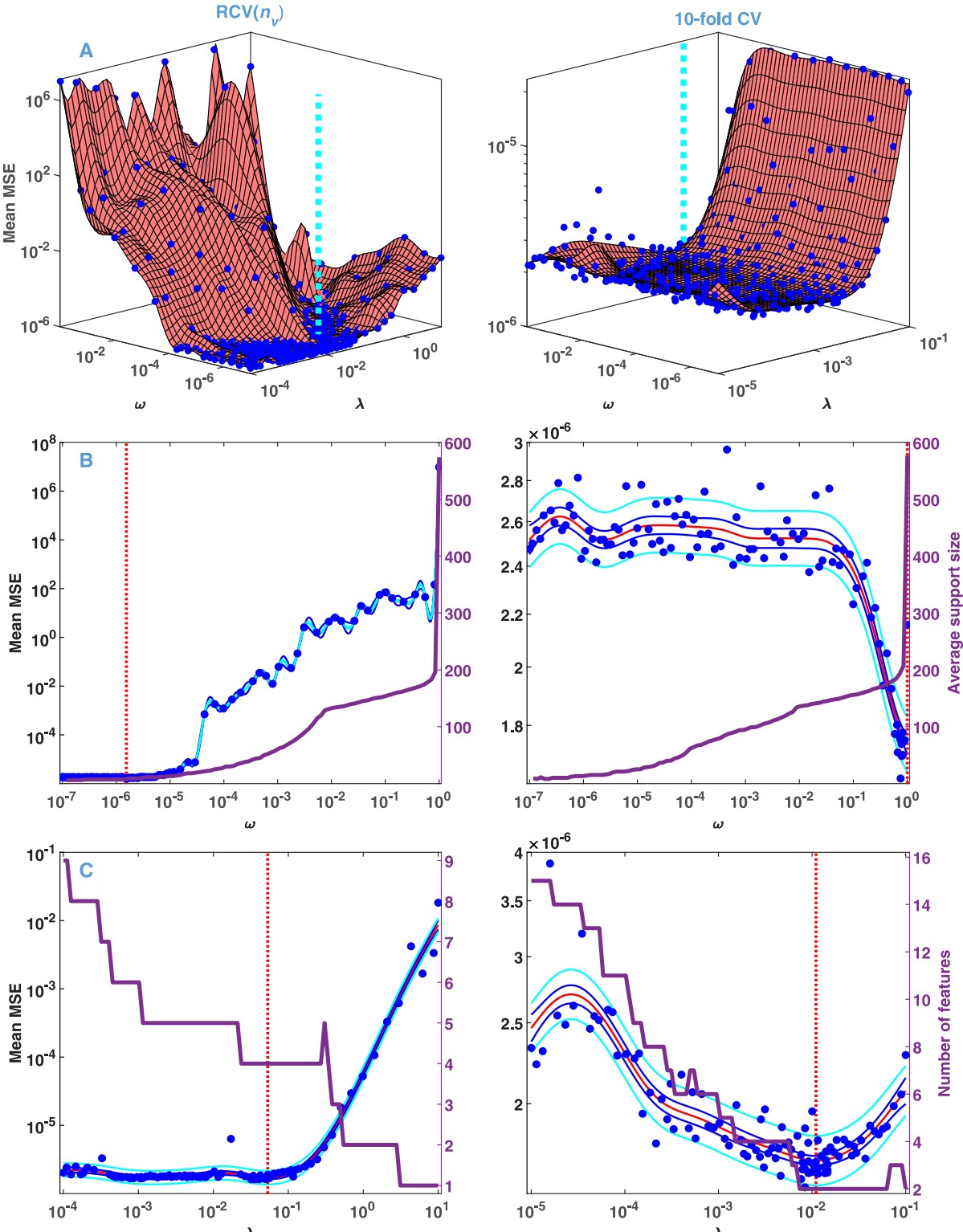

**Fig 9. Model selection based on RCV(nv) and 10-fold CV from the distributed kinetic data presented in** Fig 8. Noise level: $\sigma_{rel} = 10^{-3}$. For details see the legends and caption of Fig 3.

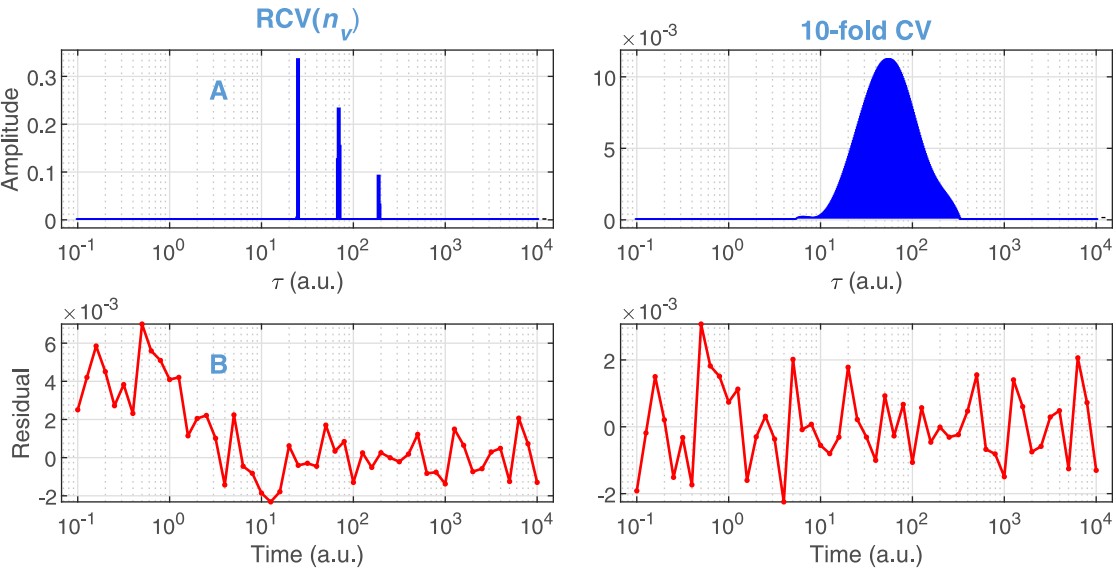

**Fig 10. Solution of GENP on the selected models based on RCV(nv) and 10-fold CV presented in Fig 9.** (A) Distribution of time constants. (B) The residual of the fits calculated with the distributions presented in (A).

Fig 12 for noise level $\sigma_{rel} = 10^{-7}$ and in S7 Fig for $\sigma_{rel} = 10^{-3}$. Strikingly, at low noise, the level of the regularization with both types of CVs is low (very low $\lambda$ and medium $\omega$), which leads to a relatively sparse solution. At the lowest regularization (10-fold CV, Fig 12B), the solution is as expected for scheme (13): a negative spike followed by a positive one with some ringing at both sides, and this characteristics remains with RCV($n_v$) (Fig 12A). The inclusion of pure exponential terms (Fig 12C and 12D) results in the expected pattern. At high noise, as anticipated, regularization is higher, but the overall pattern remains unchanged (S7 Fig). The value

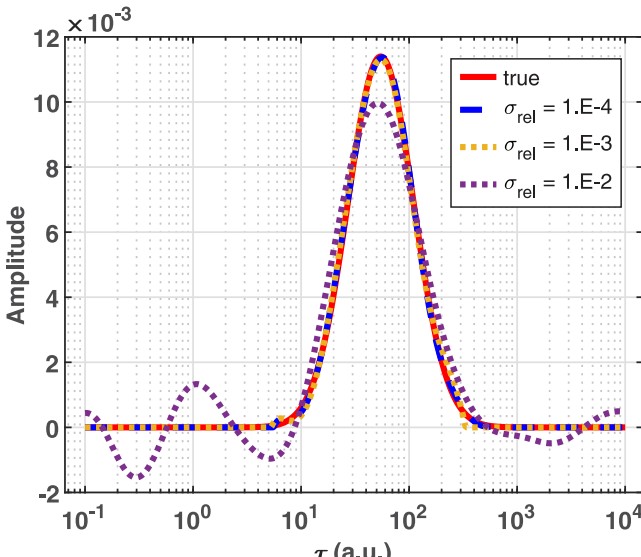

**Fig 11. Comparison of the true distribution (copied from S3B Fig) to those predicted by the selected models.** The model selection by BO based on 10-fold CV (Fig 9 right column) was carried out on the simulated data (Fig 8 blue) at different noise levels.

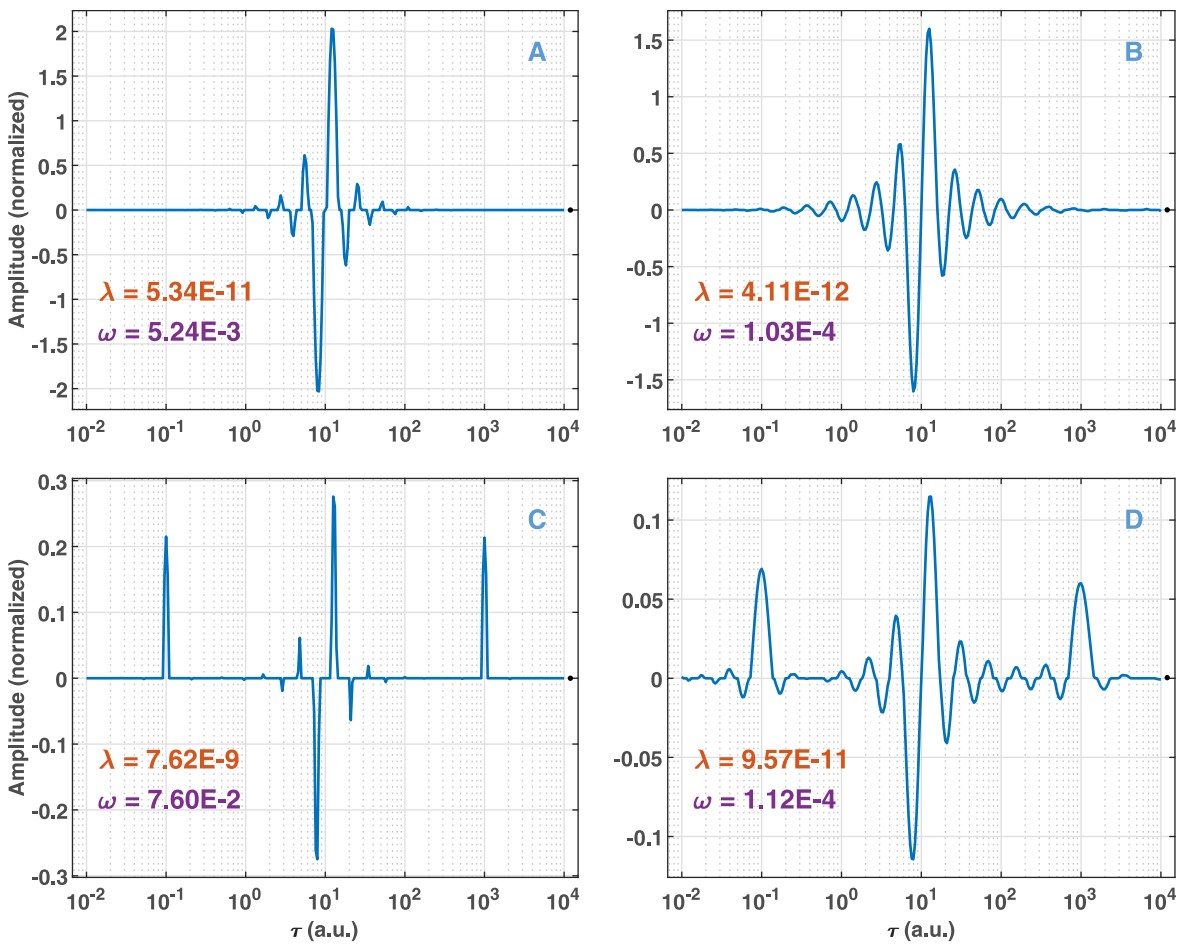

**Fig 12. Solution of the GENP obtained from data described by formula (14) at low noise.** For (A) and (C), model selection was carried out by RCV($n_v$), while for (B) and (D) by 10-fold CV. For (A) and (B), the value of $A$ in the formula is 0, while for (C) and (D) its value is 20. $\tau_1 = 10^1$, $\tau_2 = 10^{-1}$, $\tau_3 = 10^3$. Noise level: $\sigma_{rel} = 10^{-7}$. $\lambda$ and $\omega$ are the hyperparameters found by the BO process.

of $\omega$ in no case approaches 1, indicating that Algorithm 3 correctly recognizes formula (14) as first-order kinetics.

Other interesting directions of potential extensions are kinetic networks containing both first-order and other type of reactions. As a pilot study, we tested Algorithm 3 with simulated data of pure second-order kinetics and their mixture with first-order ones described by

$$\frac{C}{1 + C\frac{t}{\tau_1}} + A\left(e^{-\frac{t}{\tau_2}} + e^{-\frac{t}{\tau_3}}\right), \tag{15}$$

again with zero and nonzero value of $A$ in the presence of low and high noise levels. The solutions of the GENP with hyperparameters selected by RCV($n_v$) and 10-fold CV are presented in Fig 13 for noise level $\sigma_{rel} = 10^{-7}$ and in S8 Fig for $\sigma_{rel} = 10^{-3}$. As expected, 10-fold CV resulted in a value of $\omega$ close to 1 in every case, indicating that the overall kinetics are not of first order (Fig 13 and S8B and S8D Fig). At low noise, the presence of the included exponential terms are manifested in narrow spikes, and the complete distribution clearly indicates the mixture of first-order and non-first-order reactions (Fig 13D). At high noise, this pattern is less obvious (S8D Fig). Although the RCV($n_v$) model selection resulted in a low value of $\omega$, at low noise even this procedure provides similar patterns.

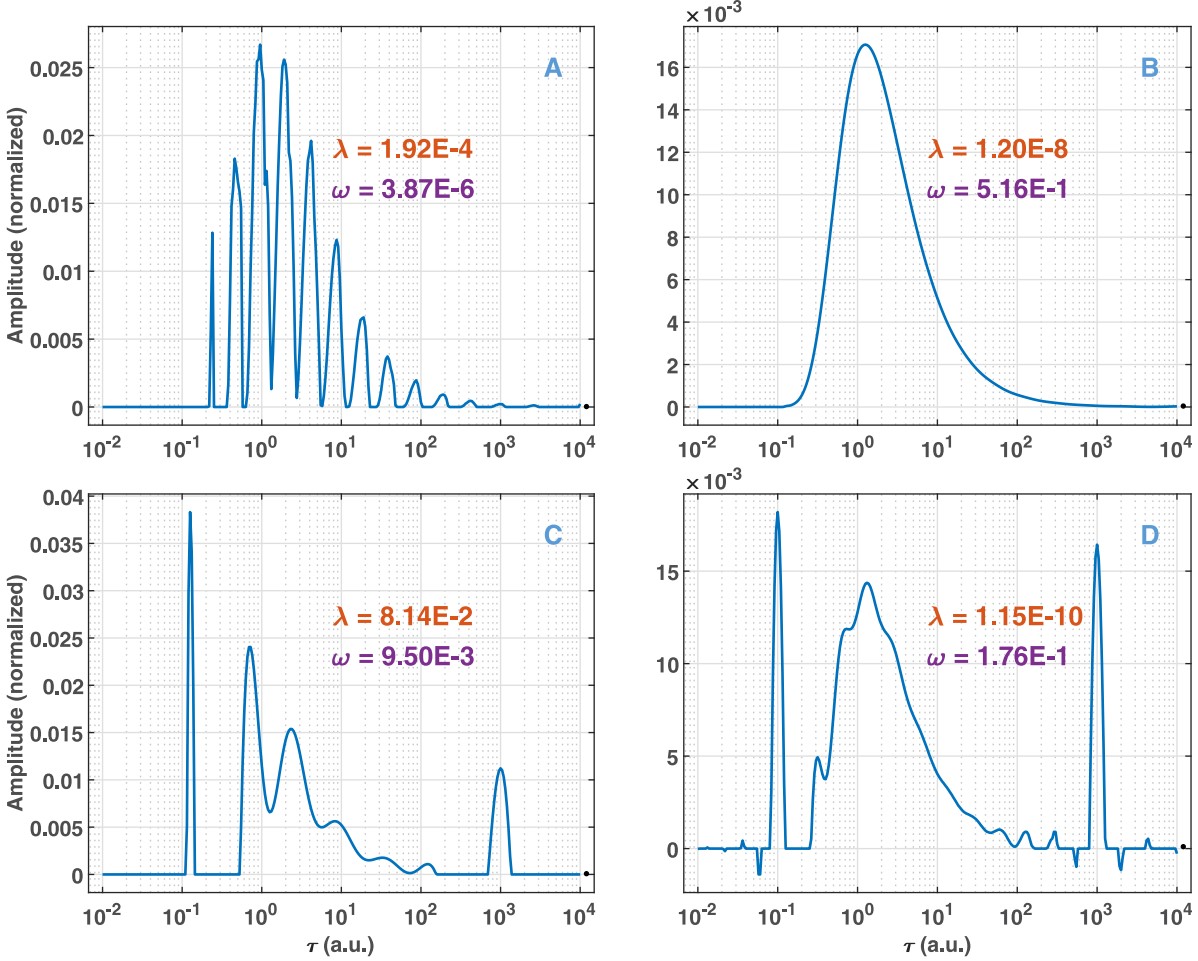

**Fig 13. Solution of the GENP obtained from data described by formula (15) at low noise.** For (A) and (C), the model selection was carried out by RCV($n_v$), while for (B) and (D) by 10-fold CV. For (A) and (B), the value of $A$ in the formula is 0, while for (C) and (D) its value is 1. $C = 8$, $\tau_1 = 10^1$, $\tau_2 = 10^{-1}$, $\tau_3 = 10^3$. Noise level: $\sigma_{rel} = 10^{-7}$. $\lambda$ and $\omega$ are the hyperparameters found by the BO process.

## Conclusions

We find that group elastic net is a powerful and flexible method for the analysis of kinetic data. With properly selected hyperparameters $\lambda$ and $\omega$ it provides a sparse recovery of kinetic parameters describing a system of first-order reactions, even of a complex scheme. Other values of the hyperparameters result in dense solutions, clearly indicating that the reaction is not of first order. We also find that the proper values of $\lambda$ and $\omega$ can be found automatically by a machine-learning algorithm, utilizing a combination of the classical $k$-fold cross-validation and its novel RCV($n_v$) version. Bayesian optimization has proved to be an ideal tool for solving the corresponding minimization problem. Our pilot study on the mixture of first-order and second-order reactions indicates that—at least at a low noise level—the presence of that mixture can be clearly recognized using Algorithm 3.

In this study we applied sophisticated statistical methods to build the machine-learning algorithms from the point of view of the experimentalist, irrespective of whether the strict theoretical conditions are satisfied for their justified application. For example, in the derivation of RCV($n_v$), Feng and Yu [48] supposed an additively separable penalty, while the group lasso

term of GENP we applied does not have this property. Similarly, the same authors supposed that the support size of the solution is less than the number of data points, which was clearly not so in our calculations with a Gaussian true distribution. Nonetheless, the applied formulae worked well in practice. Yet, further theoretical studies are needed to theoretically justify their application.

Note that both the simulated and the experimental data applied in this study were obtained from the kinetics of a light-induced process. In fact, the most complex schemes of first-order reactions can be initiated by ultrashort light pulses on physical, chemical and biological systems. Typically, the initial steps of these schemes take place in electronic and vibrational excited states, and are frequently followed by relaxation phenomena in the ground state. The demand for the algorithms presented in this study is expected to soar due to the emerging ultrafast multidimensional infrared and electronic spectroscopic techniques [13, 74], as they are able to provide large and complex datasets. The ability of our algorithm to automatically recognize nonexponential decays could be utilized in the studies of nonadiabatic systems, where the relaxation of the electronically excited state is strongly coupled to coherent nuclear wavepacket motions [75].

In addition to the analysis of experimental data, the presented methods can be used to design experiments aimed at proving hypothetical models. Simulations with different noise levels—as presented on bR (S2 Table)–can predict the signal-to-noise ratio required to demonstrate the existence of non-dominating kinetic components.

## Supporting information

**S1 Fig. Spectral and kinetic properties of the components presented in S1 Table.** (A) Absorption spectra (B) Difference spectra obtained by subtracting the spectrum of the bR state. (C) The corresponding kinetics calculated from the microscopic rate constants.
(TIF)

**S2 Fig. Simulation of realistic, wavelength- and time-dependent noise for an absorption kinetic experiment on a bR sample.** (A) Blue line: Spectral distribution of the measuring beam filtered by the bR sample. Red line: the distribution of the noise level of the blue intensity spectrum, proportional to the inverse of its square root. (B) The level of noise in the different time segments at a selected wavelength. (See Methods of the main text for details).
(TIF)

**S3 Fig. Construction of kinetic data based on the Arrhenius equation with distributed activation energy.** (A) The supposed rate constant (red) and distribution (blue) over the activation energy. (B) The true distribution of the time constant calculated from the data presented in (A). The resulted kinetics is presented in Fig 8 of the main text (blue line).
(TIF)

**S4 Fig. Detailed results of Algorithm 2 on the simulated data presented in Fig 4 of the main text.** Wavelength: 416 nm. (A) Distribution over time constants. (B) Simulated kinetics (blue) and fit (red) by the distribution presented in (A). (C) Residual of the fit presented in (B). (D) Residual of the correcting exponential fit with 8 components remaining after discretization of the data, neglecting the components with amplitudes less than 5% of the maximum amplitude value. (E) Residual of the correcting exponential fit by keeping all 17 components obtained by the discretization.
(TIF)

**S5 Fig. DADS obtained from Algorithm 2 by keeping all 17 components of the discretization.** Noise level: $\sigma_{rel} = 10^{-7}$.
(TIF)

**S6 Fig. Detailed results of Algorithm 2 on the experimental FAD fluorescence kinetic data presented in Fig 7 of the main text.** Wavelength: 520 nm. (A) Distribution over time constants. (B) Experimental kinetics (blue) and fit (red) by the distribution presented in (A). (C) Residual of the fit presented in (B). (D) Residual of the correcting exponential fit.
(TIF)

**S7 Fig. Solution of the GENP obtained from data described by formula (14) of the main text at high noise.** Noise level: $\sigma_{rel} = 10^{-3}$. For details see the caption of Fig 12 of the main text.
(TIF)

**S8 Fig. Solution of the GENP obtained from data described by formula (15) at high noise.** Noise level: $\sigma_{rel} = 10^{-3}$. For details see the caption of Fig 13 of the main text.
(TIF)

**S1 Table. Matrix of first-order microscopic rate constants corresponding to the bR photocycle model presented in Scheme 2 of the main text.**
(PDF)

**S2 Table. The predicted time constants (τ) and amplitudes (A) obtained at different noise levels, compared to the true values.** The prediction was calculated by Algorithm 1 on the simulated data at different noise levels ($\sigma_{rel}$). λ and ω are the selected hyperparameters, MSE refers to the mean square error of the fit.
(PDF)

**S3 Table. The predicted time constants (τ) and amplitudes (A) obtained with realistic wavelength- and time-dependent noise compared to the true values.** For details see the legend of S2 Table.
(PDF)

## Author Contributions

**Conceptualization:** Géza I. Groma.

**Data curation:** Géza I. Groma.

**Formal analysis:** László Zimányi, Áron Sipos, Ferenc Sarlós, Géza I. Groma.

**Funding acquisition:** László Zimányi, Géza I. Groma.

**Investigation:** Áron Sipos, Ferenc Sarlós, Rita Nagypál.

**Methodology:** László Zimányi, Áron Sipos, Ferenc Sarlós, Géza I. Groma.

**Software:** Géza I. Groma.

**Supervision:** László Zimányi, Géza I. Groma.

**Validation:** Géza I. Groma.

**Visualization:** László Zimányi, Áron Sipos, Géza I. Groma.

**Writing – original draft:** László Zimányi, Géza I. Groma.

**Writing – review & editing:** László Zimányi, Áron Sipos, Géza I. Groma.

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
