## [Decision Letter · Decision Letter 0]

8 Apr 2021

PONE-D-21-05497

Machine learning-based model selection and parameter estimation from kinetic data of complex first-order reaction systems

PLOS ONE

Dear Dr. Groma,

Thank you for submitting your manuscript to PLOS ONE. After careful consideration, we feel that it has merit but does not fully meet PLOS ONE’s publication criteria as it currently stands. Therefore, we invite you to submit a revised version of the manuscript that addresses the points raised during the review process.

We look forward to receiving your revised manuscript.

Kind regards,

Ramon Grima

Academic Editor

PLOS ONE

Journal Requirements:

1. Please ensure that your manuscript meets PLOS ONE's style requirements, including those for file naming. The PLOS ONE style templates can be found athttps://journals.plos.org/plosone/s/file?id=wjVg/PLOSOne_formatting_sample_main_body.pdf and https://journals.plos.org/plosone/s/file?id=ba62/PLOSOne_formatting_sample_title_authors_affiliations.pdf

Additional Editor Comments (if provided):

Reviewers' comments:

Reviewer's Responses to Questions

**Comments to the Author**

1. Is the manuscript technically sound, and do the data support the conclusions?

Reviewer #1: Yes

Reviewer #2: Yes

2. Has the statistical analysis been performed appropriately and rigorously? 

Reviewer #1: Yes

Reviewer #2: Yes

3. Have the authors made all data underlying the findings in their manuscript fully available?

Reviewer #1: Yes

Reviewer #2: Yes

4. Is the manuscript presented in an intelligible fashion and written in standard English?

Reviewer #1: Yes

Reviewer #2: No

5. Review Comments to the Author

Reviewer #1: In this paper the authors use sparse modelling techniques in order to infer rate parameters for first-order reaction networks. My recommendation is that this article is accepted with minor revisions. The paper argues its case very well, and they carefully lay-out the LASSO-type sparse regression procedure used in order to infer the model parameters. In particular, they take great care in optimising the hyperparameters of the LASSO procedure via Bayesian Optimisation, whereas often in other publications the LASSO parameters are ascertained empirically. Additionally, I was impressed by the quality of the figures in the paper which are stylish, colorful and easy to read. However, I do have some issues to raise that I would like the authors to respond to:

1. The introduction gives a too brief overview of the problem the authors are attempting to solve. They could add a schematic figure detailing exactly what question they are intending to answer, and the workflow of their intended procedure. Personally, I am unfamiliar with DADS and DAS and would have appreciated a more detailed introduction to them, which would help keep the paper self-contained. Also, to aid experimentalists you could detail exactly what you mean by the microscopic rate constants/matrix with respect to a simple reaction system for which you can write the deterministic rate equations explicitly.

2. Although not awful, the written English could certainly benefit from a thorough proof-reading as often sentences are awkwardly constructed.

3. Even though I really like the presentation of the figures the captions need to be much more detailed since it is sometimes unclear exactly what is being shown. For example, although they attempted to explain in the text, I am unsure of the meaning behind Fig 2B and D (and similar figures). I understand that they relate to the sparsity and amplitude of the basis functions but I am unsure exactly what is being shown to draw their conclusions from. Another example is the caption of Fig 3, where the authors need to write at least a sentence of explanation to accompany sub-figure and aid the narrative (currently the entire caption comprises around 10 words). Same comments can be made for the other figure captions.

4. The method is shown to (almost) correctly infer the number of components needed for their example model (Scheme 1) and additionally to infer the correct kinetic parameters from simulated data (Table 1). However, surely it would be the case that experimentalists do not know a priori whether their system is indeed made up solely of first-order reactions. Although the authors do make hints to future work dealing with non-first-order kinetics in the conclusion, as it stands the method seems to have little use where in reality reaction systems may contain bimoleular reactions, time dependent rate constants or non-exponential waiting time distributions. I would like the authors to show the outputs of their sparse regression procedure when applied to reaction networks including non-first-order kinetics. Interesting questions in the analysis of this include: (i) in reaction systems no solely consisting of first order reactions, are the reactions that are of first order correctly inferred?, (ii) for a reaction with a non-exponential waiting time distribution, say an Erlang or Hypoexponential distributed waiting time distribution, does the sparse regression procedure manage to effective infer the effective number of additional number of species introduced by these waiting time distributions and the associated hyperparameters? It is very important to address such questions, since if the sparse regression procedure works only for reaction systems we know to be first order a priori it could not be used on experimental data for which we do not know the underlying details.

5. The inference of the kinetic parameters is shown to be relatively robust with respect to the artificial noise added to the experimental data (Table 1). However, the choice of this ‘noise size’ is arbitrary, and a much better test of the robustness of the procedure to realistic stochastic noise would be to include it individually on each species, being specified by the linear noise approximation (LNA). If the authors are unfamiliar, the LNA gives an estimate for the intrinsic noise based on an expansion of the master equation in the system size, keeping only the terms to order Ω-1/2. Read “van Kampen, Stochastic Processes in Physics and Chemistry, Chapter X” for more details.

Reviewer #2: Please see my PDF review and the scanned manuscript. My main contribution is to help with the English grammar in the authors' description. I specified "Minor revision", but a serious effort is needed to smooth the paper's readability.

6. PLOS authors have the option to publish the peer review history of their article (what does this mean?). If published, this will include your full peer review and any attached files.

Reviewer #1: **Yes: **James Holehouse

Reviewer #2: **Yes: **Michael A Saunders

---

## [Author Response · Author response to Decision Letter 0]

22 Jun 2021

Reviewer #1:

1. The introduction gives a too brief overview of the problem the authors are attempting to solve. They could add a schematic figure detailing exactly what question they are intending to answer, and the workflow of their intended procedure. Personally, I am unfamiliar with DADS and DAS and would have appreciated a more detailed introduction to them, which would help keep the paper self-contained. Also, to aid experimentalists you could detail exactly what you mean by the microscopic rate constants/matrix with respect to a simple reaction system for which you can write the deterministic rate equations explicitly.

Author’s reply:

A figure (Scheme 1) showing the workflow of the paper is added to the last paragraph of the Introduction, which narratively explains the aims of the paper. The Introduction is also extended with a paragraph, explicitly stating the system of differential equations the first-order reactions are based on. The same paragraph defines the terms of microscopic and macroscopic rate constants as well as DAS and DADS. These terms are widely used in the literature of first-order kinetics but, indeed, the names microscopic/macroscopic are somewhat misleading as they have nothing to do with the physical size of the system.

Reviewer #1:

2. Although not awful, the written English could certainly benefit from a thorough proof-reading as often sentences are awkwardly constructed.

Author’s reply:

The revised text has gone through a massive grammatical and syntactical editing.

Reviewer #1:

3. Even though I really like the presentation of the figures the captions need to be much more detailed since it is sometimes unclear exactly what is being shown. For example, although they attempted to explain in the text, I am unsure of the meaning behind Fig 2B and D (and similar figures). I understand that they relate to the sparsity and amplitude of the basis functions but I am unsure exactly what is being shown to draw their conclusions from. Another example is the caption of Fig 3, where the authors need to write at least a sentence of explanation to accompany sub-figure and aid the narrative (currently the entire caption comprises around 10 words). Same comments can be made for the other figure captions.

Author’s reply:

The figure legends are substantially extended, especially at the suggested points.

Reviewer #1:

4. The method is shown to (almost) correctly infer the number of components needed for their example model (Scheme 1) and additionally to infer the correct kinetic parameters from simulated data (Table 1). However, surely it would be the case that experimentalists do not know a priori whether their system is indeed made up solely of first-order reactions. Although the authors do make hints to future work dealing with non-first-order kinetics in the conclusion, as it stands the method seems to have little use where in reality reaction systems may contain bimoleular reactions, time dependent rate constants or non-exponential waiting time distributions. I would like the authors to show the outputs of their sparse regression procedure when applied to reaction networks including non-first-order kinetics. Interesting questions in the analysis of this include: (i) in reaction systems no solely consisting of first order reactions, are the reactions that are of first order correctly inferred?, (ii) for a reaction with a non-exponential waiting time distribution, say an Erlang or Hypoexponential distributed waiting time distribution, does the sparse regression procedure manage to effective infer the effective number of additional number of species introduced by these waiting time distributions and the associated hyperparameters? It is very important to address such questions, since if the sparse regression procedure works only for reaction systems we know to be first order a priori it could not be used on experimental data for which we do not know the underlying details.

Author’s reply:

The Reviewer raised interesting problems in relation to the extension of our studies. Especially, formulae identical to those of the Erlang distributions also occur in the solution of first-order reactions if the K matrix has degenerate eigenvalue(s). Following his suggestion, we tested Algorithm 3 for a simulated kinetics of the form t*exp(-t/tau), both alone and in combination with pure exponential terms. We found that the algorithm correctly recognized that the kinetics is of first order (Fig 12 and S7 Fig). In addition, simulated second-order kinetics both alone and in combination with exponential terms were correctly recognized as of not first-order (Fig 13 and S8 Fig). At least at low noise levels, in the mixture retains the dense and sparse characters of the second-order and first-order components, respectively (Fig 13 C and D).

Reviewer #1:

5. The inference of the kinetic parameters is shown to be relatively robust with respect to the artificial noise added to the experimental data (Table 1). However, the choice of this ‘noise size’ is arbitrary, and a much better test of the robustness of the procedure to realistic stochastic noise would be to include it individually on each species, being specified by the linear noise approximation (LNA). If the authors are unfamiliar, the LNA gives an estimate for the intrinsic noise based on an expansion of the master equation in the system size, keeping only the terms to order Ω-1/2. Read “van Kampen, Stochastic Processes in Physics and Chemistry, Chapter X” for more details.

Author’s reply:

We agree with the Reviewer in that the independent and identical distribution (iid) property of noise supposed in the original version of the manuscript often remains unsatisfied in real experiments. As we argue in the revised version, however, the suggested consideration of the intrinsic noise due to the small size of the reaction system in the framework of NLA is impossible for real experimental data on the kinetics of the bR photocycle for which our simulations are presented. This is due to the fact that the noise level originating from a realistic measuring procedure is by more than four orders of magnitude higher. On the other hand, as shown in detail in the revised manuscript, in a real experiment, for technical reasons, both the temporal and spectral distribution of the noise highly violate the iid property (S2 Fig). According to our results presented in the revised manuscript, this violation does not have any effect on the performance of Algorithm 3 (S3 Table).

Reviewer #2:

Please see my PDF review and the scanned manuscript. My main contribution is to help with the English

grammar in the authors' description. I specified "Minor revision", but a serious effort is needed to smooth the paper's readability.

Author reply:

We highly appreciate the Reviewer’s help in improving the readability of the paper.

---

## [Decision Letter · Decision Letter 1]

20 Jul 2021

PONE-D-21-05497R1

Machine-learning model selection and parameter estimation from kinetic data of complex first-order reaction systems

PLOS ONE

Dear Dr. Groma,

Thank you for submitting your manuscript to PLOS ONE. After careful consideration, we feel that it has merit but does not fully meet PLOS ONE’s publication criteria as it currently stands. Therefore, we invite you to submit a revised version of the manuscript that addresses the points raised during the review process.

We look forward to receiving your revised manuscript.

Kind regards,

Ramon Grima

Academic Editor

PLOS ONE

Journal Requirements:

Reviewers' comments:

Reviewer's Responses to Questions

**Comments to the Author**

1. If the authors have adequately addressed your comments raised in a previous round of review and you feel that this manuscript is now acceptable for publication, you may indicate that here to bypass the “Comments to the Author” section, enter your conflict of interest statement in the “Confidential to Editor” section, and submit your "Accept" recommendation.

Reviewer #1: All comments have been addressed

Reviewer #2: All comments have been addressed

2. Is the manuscript technically sound, and do the data support the conclusions?

Reviewer #1: Yes

Reviewer #2: Yes

3. Has the statistical analysis been performed appropriately and rigorously? 

Reviewer #1: Yes

Reviewer #2: Yes

4. Have the authors made all data underlying the findings in their manuscript fully available?

Reviewer #1: Yes

Reviewer #2: Yes

5. Is the manuscript presented in an intelligible fashion and written in standard English?

Reviewer #1: Yes

Reviewer #2: Yes

6. Review Comments to the Author

Reviewer #1: The authors have addressed all of the concerns that I previously raised and the document is now much more improved. In particular the authors have put a lot of effort in addressing the query I had if their procedure is applied to systems with non-first order reactions - doing so with two examples. There are still two minor concerns that hopefully the authors can change before publication:

1. In the new Section from line 739-751 there is a conflicting notation in "A" which is used both as a species in (13) and as an algebraic parameter in Eq (12) - please change one of these.

2. The authors state that in response to query 5 that "As we argue in the revised version, however, the suggested consideration of the intrinsic noise due to the small size of the reaction system in the framework of NLA [sic] is impossible for real experimental data on the kinetics of the bR photocycle for which our simulations are presented. This is due to the fact

that the noise level originating from a realistic measuring procedure is by more than four orders of magnitude higher". Could the authors please provide references in support of this statement?

Reviewer #2: The original manuscript required many grammatical corrections as indicated with my previous review.

The authors have dealt with these diligently. Just a few minor corrections are needed in the revision.

7. PLOS authors have the option to publish the peer review history of their article (what does this mean?). If published, this will include your full peer review and any attached files.

Reviewer #1: **Yes: **James Holehouse

Reviewer #2: **Yes: **Michael Alan Saunders

---

## [Author Response · Author response to Decision Letter 1]

21 Jul 2021

Reviewer #1:

1. In the new Section from line 739-751 there is a conflicting notation in "A" which is used both as a species in (13) and as an algebraic parameter in Eq (12) - please change one of these.

Author’s reply:

The notations in (13) were changed.

Reviewer #1:

2. The authors state that in response to query 5 that "As we argue in the revised version, however, the suggested consideration of the intrinsic noise due to the small size of the reaction system in the framework of NLA [sic] is impossible for real experimental data on the kinetics of the bR photocycle for which our simulations are presented. This is due to the fact that the noise level originating from a realistic measuring procedure is by more than four orders of magnitude higher". Could the authors please provide references in support of this statement?

Author’s reply:

The requested references were added in line 602.

Reviewer #2:

The original manuscript required many grammatical corrections as indicated with my previous review.

The authors have dealt with these diligently. Just a few minor corrections are needed in the revision.

Author reply:

Many thanks for the further corrections of the Reviewer, all of them were accepted, and the manuscript text was amended accordingly.

---

## [Editor Report · Decision Letter 2]

22 Jul 2021

Machine-learning model selection and parameter estimation from kinetic data of complex first-order reaction systems

PONE-D-21-05497R2

Dear Dr. Groma,

We’re pleased to inform you that your manuscript has been judged scientifically suitable for publication and will be formally accepted for publication once it meets all outstanding technical requirements.

Kind regards,

Ramon Grima

Academic Editor

PLOS ONE
---

## [Editor Report · Acceptance letter]

27 Jul 2021

PONE-D-21-05497R2 

Machine-learning model selection and parameter estimation from kinetic data of complex first-order reaction systems 

Dear Dr. Groma:

I'm pleased to inform you that your manuscript has been deemed suitable for publication in PLOS ONE. Congratulations! Your manuscript is now with our production department. 

Kind regards, 

on behalf of

Prof. Ramon Grima 

Academic Editor

PLOS ONE